# Progenitor translatome changes coordinated by *Tsc1* increase perception of Wnt signals to end nephrogenesis

Alison E. Jarmas [1,2,4], Eric W. Brunskill[1,2,4], Praneet Chaturvedi[1,2], Nathan Salomonis [3] & Raphael Kopan [1,2 ✉]

Mammalian nephron endowment is determined by the coordinated cessation of nephrogenesis in independent niches. Here we report that translatome analysis in $Tsc1^{+/-}$ nephron progenitor cells from mice with elevated nephron numbers reveals how differential translation of Wnt antagonists over agonists tips the balance between self-renewal and differentiation. Wnt agonists are poorly translated in young niches, resulting in an environment with low R-spondin and high Fgf20 promoting self-renewal. In older niches we find increased translation of Wnt agonists, including R-spondin and the signalosome-promoting Tmem59, and low Fgf20, promoting differentiation. This suggests that the tipping point for nephron progenitor exit from the niche is controlled by the gradual increase in stability and possibly clustering of Wnt/Fzd complexes in individual cells, enhancing the response to ureteric bud-derived Wnt9b inputs and driving synchronized differentiation. As predicted by these findings, removing one *Rspo3* allele in nephron progenitors delays cessation and increases nephron numbers in vivo.

[1] Department of Pediatrics, University of Cincinnati College of Medicine, Cincinnati, OH, USA. [2] Division of Developmental Biology, Cincinnati Children's Hospital Medical Center, Cincinnati, OH, USA. [3] Division of Biomedical Informatics, Cincinnati Children's Hospital Medical Center, Cincinnati, OH, USA. [4] These authors contributed equally: Alison E. Jarmas, Eric W. Brunskill. ✉email: Raphael.Kopan@cchmc.org

Nephron number varies in humans 10-fold, ranging from $2 \times 10^5$ to $2.5 \times 10^6$ filtration units per kidney[1–3], compared to two-fold differences observed between different mouse strains (Supplementary Fig. S1A, D). Individuals with low nephron endowment are at high risk for developing hypertension, chronic kidney disease (CKD), and end-stage renal disease (ESRD)[4–13]. Infants born prior to 30 weeks gestational age undergo postnatal nephrogenesis for at most 40 days after birth[12] and some or most nephrons formed after preterm birth appear abnormal[13] and may not function properly. Currently, neonates born at 24 weeks have a survival rate of over 60%[14], and are predicted to be at the extreme low end of nephron endowment and at high risk for early-onset CKD and ESRD as compared to individuals born at full-term[15–18]. With the improvements in survival of extremely low-birth-weight and preterm infants, the negative implications of low nephron endowment and the associated economic and quality of life impacts will increase[18]. Understanding the mechanism(s) that control the synchronous cessation of nephrogenesis could be leveraged to develop interventions aimed at extending the process and increasing nephron endowment in at-risk populations, including premature and low-birth-weight infants.

Mammalian nephrogenesis depends on transiently amplifying nephron progenitor cells (NPCs) localized in a tight cap on the distal aspect of ureteric bud (UB) tips[19,20]. The NPCs are sustained by juxtacrine Fgf20/9 signaling and reciprocal interactions between the NPCs and the UB, which secretes Wnt9b that binds to Fzd (Frizzled) receptors on NPCs and, in a process involving numerous molecular components, stabilizes ß-catenin (the canonical pathway, reviewed in[21,22]). ß-Catenin enters the nucleus where it will interact with Tcf/Lef family of transcription factors to regulate gene expression promoting both self-renewal and differentiation of NPC[23–27]. Lrp co-receptors, members of the Lgr family, secreted R-spondins and membrane-localized E3 ubiquitin ligases[21,22] all regulate the duration/strength of the signal; alterations in these factors could tune NPC responses to canonical Wnt input. Further, the balance between self-renewal and differentiation is regulated in part by stromal signals that may include Fat4[28] and Decorin[29]. Unlike the mesonephric kidney, which can add nephrons throughout adult life via long-lived stem cells[30], the mammalian metanephric kidney has a finite window during which nephrons are generated. In humans, NPCs are exhausted en masse by 34–36 weeks of gestation, with 60% of nephrons forming during the third trimester[31]. In mice, nephrogenesis ends by postnatal day 3 (P3)[32,33]. Historically, the NPC population has been described as a transitory cell population that progresses from an uncommitted or naïve state (*Six2+; Cited1+*), to a committed, primed (*Six2+; Cited1−; Wnt4+*) state, to pretubular aggregates (PTA, *Six2+; Cited1−; Wnt4+, Pax8+; Fgf8+*) that will ultimately complete a mesenchymal-to-epithelial transition (MET) to form each nephron. However, this linear progression toward epithelium has been recently challenged with the observation that *Six2+; Cited1−; Wnt4+* cells can migrate and revert back to the naïve, *Six2+; Cited1+, Wnt4−* state[34].

Cessation of nephrogenesis is not due to an extrinsic trigger (e.g., loss of niche factors[35]), to an intrinsic age-dependent change(s) affecting the progenitors' ability to self-renew (e.g.[36]), or to an altered cap mesenchyme (CM)/UB ratio[37]. Instead, heterochronic engraftment experiments demonstrated that transplanted, older naïve mouse NPCs engraft in a young niche better than age-matched committed NPCs but more poorly than young, committed NPCs. Those that engraft and remain in the niche are surrounded by young, FGF20-expressing NPCs and can contribute nephrons for up to twice the normal lifespan of a mouse NPC in situ[38]. These findings led us to propose a tipping-point model controlling cessation: when most neighboring cells display a young signature (which includes Fgf20, a niche-retention signal[38,39]), NPCs gaining an old transcriptome signature persist in the niche. When the number of old cells among immediate neighbors rises above a hypothetical threshold, the model assumes all remaining progenitors will differentiate. *Tsc1* (hamartin) has emerged as one gene that might regulate the rate of progression to the tipping point, or the interpretation of the exit signal, since mice with NPCs hemizygous for this gene have delayed cessation and produce 25% more nephrons[40]. Despite these insights, the precise signaling mechanism(s) that drive the coordinated exit of NPCs from all nephrogenic niches at the time of cessation are not completely understood.

In this work, to identify molecular mechanism(s) controlling cessation of nephrogenesis, we leverage two existing models with increased nephron endowment—*Six2-CreERT2* (*Six2^CE/+^*, herein *Six2^KI^*[19,41]) and *Six2^TGC^;Tsc1^+/Flox^* (*Six2^TGC;Tsc1^*[40])—and analyze the transcriptome at two developmental time points (embryonic day 14 (E14) and postnatal day 0 (P0)) of three genotypes: *Six2^KI^*, *Six2^TGC;Tsc1^*, and the control *Six2^TGC^*[19]. Our analyses further establish that the two advantageous mutations enhance nephrogenesis via temporally disparate effects on the nephron progenitor population. *Six2* hemizygosity affects the young NPCs via a mechanism yet to be fully explored. The transcriptional signatures identified at P0 prompted translatome analysis of control *Six2^TGC^* and *Six2^TGC;Tsc1^* NPCs, revealing that (1) old (P0) cells enhance the translation of components in all major developmental signaling pathways, making these cells more acutely aware of their signaling environment and (2) *Tsc1* hemizygosity differentially affects the relative abundance of key Wnt pathway components in older P0 NPC, lowering translation of agonists (*Rspo1, Rspo3*, and *Tmem59*[42]), and elevating translation of some antagonists (*Kremen1, Dkk1, Znrf3/Rnf43*) to modulate the Wnt9b signals received by these cells. We note that mTOR activity, evaluated by accumulation of phosphorylated ribosomal protein S6, is elevated in *Six2^TGC^;Tsc1^Flox/Flox^* but not in *Six2^TGC;Tsc1^* NPCs relative to age-matched controls, and thus cannot explain differential translation seen in *Six2^TGC;Tsc1^*. We validate the hypothesis that change in Wnt-responsiveness impacts cessation by demonstrating in vivo that *Rspo3* hemizygosity in NPCs (shortening the half-life of Wnt/Fzd complexes) delays cessation and increases nephron endowment. These observations support the hypothesis that the tipping point is controlled by the gradual increase within individual cells in the stability of Wnt/Fzd complexes, increasing the magnitude of the response to consistent Wnt9b inputs. These findings provide a potential therapeutic framework for delaying cessation and increasing nephron endowment in vulnerable populations.

## Results

**Kidney niche composition does not change with nephron number.** Increased nephron numbers observed in different genetic models could arise as a result of common or distinct mechanisms. First, increased nephron endowment may reflect an expanded population of naïve NPCs. Second, the same or different age-dependent changes in the transcriptome[38] and/or metabolome[43] may unfold more slowly, enabling NPC to retain their youthful character relative to chronologically matched NPCs from the control strain, *Six2^TGC^*. Third, a molecular mechanism(s) unrelated to the relative age of the progenitors is involved, perhaps creating a distinct, novel population of NPC. Predictions of all three hypotheses could be tested by analyzing the transcriptomes of NPC populations isolated from mouse kidneys of each of the three genotypes (*Six2^TGC^, Six2^TGC;Tsc1^*, and *Six2^KI^*) at two developmental time points representing young and old NPC (E14 and P0). Single-cell RNA-sequencing (scRNA-Seq) profiles

generate sufficient markers to segregate even closely related cell types within tissues, enabling identification of NPC population(s) correlated with the increase in nephron numbers. Thus, we pursued scRNA-Seq to screen for gene expression changes which may provide mechanistic insight into the nephron phenotypes.

To generate six scRNA-Seq datasets (three genotypes, each at E14 and P0), we dissociated kidneys for 5–7 min at 10 °C with psychrophilic proteases[44] in an Eppendorf Thermomixer R, generated single-cell suspensions, retained only samples with >85% viability (assessed by trypan blue staining), and submitted the pooled cells from several individual animals for sequencing on the 10x Genomics Chromium v2 platform. Our samples were enriched for cells in the renal cortical region due to the rapid, peripheral digestion of the tissue. The exclusion of a fluorescence-activated cell sorting (FACS)-isolation step allowed us to (1) capture other cortical niche-residing cell types involved in nephrogenesis, and (2) reduce the potential introduction of cell stress signatures generated by flow cytometry.

In our bioinformatic analysis of these resulting datasets, we adhered to best practices[45] and implemented multiple pipelines or packages when possible to seek bioinformatic consensus. For the initial analyses, we applied Seurat (v2.3.4) to filter, reduce dimensions and cluster cells (see 'Methods' for details). Unsupervised clustering using t-distributed stochastic neighbor embedding (tSNE[46]) of the six samples individually defined between 14 and 18 transcriptionally distinct clusters, representing various cell types derived from the developing kidney cortex including nephron progenitors (NPC), ureteric bud (UB), stroma (S), renal vesicle (RV), s-shaped body (SSB), pretubular aggregate (PTA), proximal tubule (PT), and distal tubule (DT) cells (Fig. 1A, Supplementary Fig. S1C, and Supplementary Data 1).

To confirm that we captured all cortical cell types found in the developing kidney, and to evaluate how different bioinformatic parameters could potentially mask or reveal the presence of novel cell clusters, we cross-referenced our data with published mouse kidney scRNA datasets. Mutual Nearest Neighbor (FastMNN) correction[47,48] in Seurat (v3.2.3) was applied to remove batch effects and perform systematic comparisons between E14 $Six2^{KI}$ data and an age-matched existing $Six2^{KI}$ dataset[49]. Mutual nearest neighbor (MNN) information was then used to calculate the similarity among cell types from the two datasets. Of note, both sample processing and bioinformatic assumptions vary between datasets; Combes et al.[49] analyzed wild-type (E18.5) and Six2 hemizygotes (E14.5) processed via dissociation at 37 °C, performed multiple independent scRNA-Seq runs to control for batch effects, and filtered out ribosomal, mitochondrial, and lncRNA genes in their analytical pipeline (using Seurat v2.4). Importantly, while the methodological differences changed the relative rank of individual marker genes, all (and only) the cell identities assigned by Combes et al. were present in each of our samples (Supplementary Fig. S1C), confirming that no new NPC populations were present. As an additional, unbiased test we used GO-Elite (http://www.genmapp.org/go_elite/) to identify previously annotated cell types most highly correlated with similarly filtered clusters from Combes' and our data (Fig. 1B and Supplementary Data 2). This analysis correctly identified the cortical bias in our scRNA dataset (e.g., lack of mature proximal tubule cells, ductal stalk cell types) resulting from our sample preparation method, and showed that the NPC-Str cluster (Combes C14) was a minimal contributor in our data, consistent with it being a technical artifact as proposed by the authors[49]. Supplementary Data 1 contains all the markers for each of the E14 $Six2^{KI}$ clusters, with additional information about pathways and transcription factors in Supplementary Data 2; the markers characterizing each cluster are presented in Supplementary Data 3, with each sample in a separate tab.

Having validated that cortically derived cell types found in developing kidneys are appropriately represented in our data, we proceeded to investigate the hypothesis that an expanded cell population distinguished the high nephron number lines ($Six2^{KI}$ and $Six2^{TGC;Tsc1}$) from the control ($Six2^{TGC}$). We pooled all six samples and performed an integration analysis using the Seurat 3.0 UMAP algorithm[50] to visualize clusters which emphasized global similarities; annotations for barcode, cluster, and cell type designation across all samples are provided in Supplementary Data 4. This analysis identified several super clusters (continents), characterized as NPC, stromal, and ureteric bud (UB, $Ret+$) continents. The single UB cluster contained cells across all time points and genotypes (Supplementary Fig. S1E and Supplementary Data 5). Notably, UMAP correctly aligned the NPC along a proximo-distal developmental axis, with the naïve $Six2+$, $Cited1+$ cells at one pole of the cluster, followed by a transitional population ($Six2+$, $Wnt4+$), and segregating into epithelial renal vesicle and precursors of proximal tubule (PT, $Hnf4a+$), distal tubule (DT, $Tfap2b+$, $Gata3+$), and podocyte (Pod, $Mafb+$) at the opposite pole (Figs. 1Ci and 2A–D). Most of the naïve NPCs were in G1 (Figs. 1Cii and 2D). Importantly, assigning a unique color to each genotype and developmental time point reveals that unlike erythrocytes, the majority of which were contributed by the E14 $Six2^{TGC;Tsc1}$ sample (green), no sample dominated a particular area of the NPC super-cluster (Figs. 1Ciii and 2A, and Supplementary Fig. S1E, F). Only three minor clusters were not shared by all samples: cluster 20 that appears to contain neurons, cluster 17 that was designated unassigned and may contain doublets, and cluster 15, marked by $Dcn$[29] expression, which was enriched in the E14 samples (Supplementary Fig. S1G). However, this enrichment may reflect the poor recovery of medullary stroma (where $Dcn$ is expressed[49]) due to increase in kidney size at P0. We conclude that the differences between the two strains generating more nephrons and control did not reflect over-abundance of any particular NPC subtype, nor the emergence of new cell types.

**$Six2, Tsc1$ hemizygous NPCs appear older than controls**. To increase resolution and find gene expression signatures that might be masked by analyzing all six samples at once, we performed pair-wise integration analysis, first comparing E14 to P0 cells for each genotype (Fig. 2E–G) to identify patterns associated with NPC aging. In our previous report, we used the Fluidigm C1 system to establish that transcriptional signatures distinguish old (E18.5, P0) from young (E12, E14.5) NPC, and the most notable changes were increased mitochondrial and ribosomal signatures in E18.5 and P0 NPC compared to E12.5–E14 NPCs[38], which fueled our interest in $Mtor$ and $Tsc1$[40].

For each contributing NPC cluster, we visualized differential gene expression using a dot plot to show the relative expression of a subset of marker genes and the fraction of cells expressing a given gene (Fig. 2E, F). We selected genes with graded expression enriched in naïve NPC ($Cited1, Six2, Hoxc10, Wt1$), in committed or transitional NPC ($Six2, Wt1, Wnt4, Pax8$), in PTA ($Nrarp, Lhx1, Hnf1b$), and in the emerging epithelial lineages ($Epcam$, all epithelia; $Hnf4a$, proximal tubule; $Fgf8, Gata3$, distal tubules). Based on the relative expression of $Cited1$ and $Wnt4$ in the $Six2^{TGC}$ control strain (Fig. 2E), in $Six2^{KI}$ (Fig. 2F), and in $Six2^{TGC;Tsc1}$ NPC (Fig. 2G), $Tsc1^{+/−}$ NPCs are not more naïve in character (i.e., P0 cells did not resemble E14 cells). Next, we performed pair-wise comparisons of each high nephron-producing NPC (blue circles, Fig. 2H–K) to its age-matched $Six2^{TGC}$ control (red circle, Fig. 2H–K). Again, at P0, $Six2^{KI}$ and $Six2^{TGC;Tsc1}$ appeared to contribute most of the $Wnt4$-expressing cells (Fig. 2J, K and Supplementary Fig. S2). These trends are

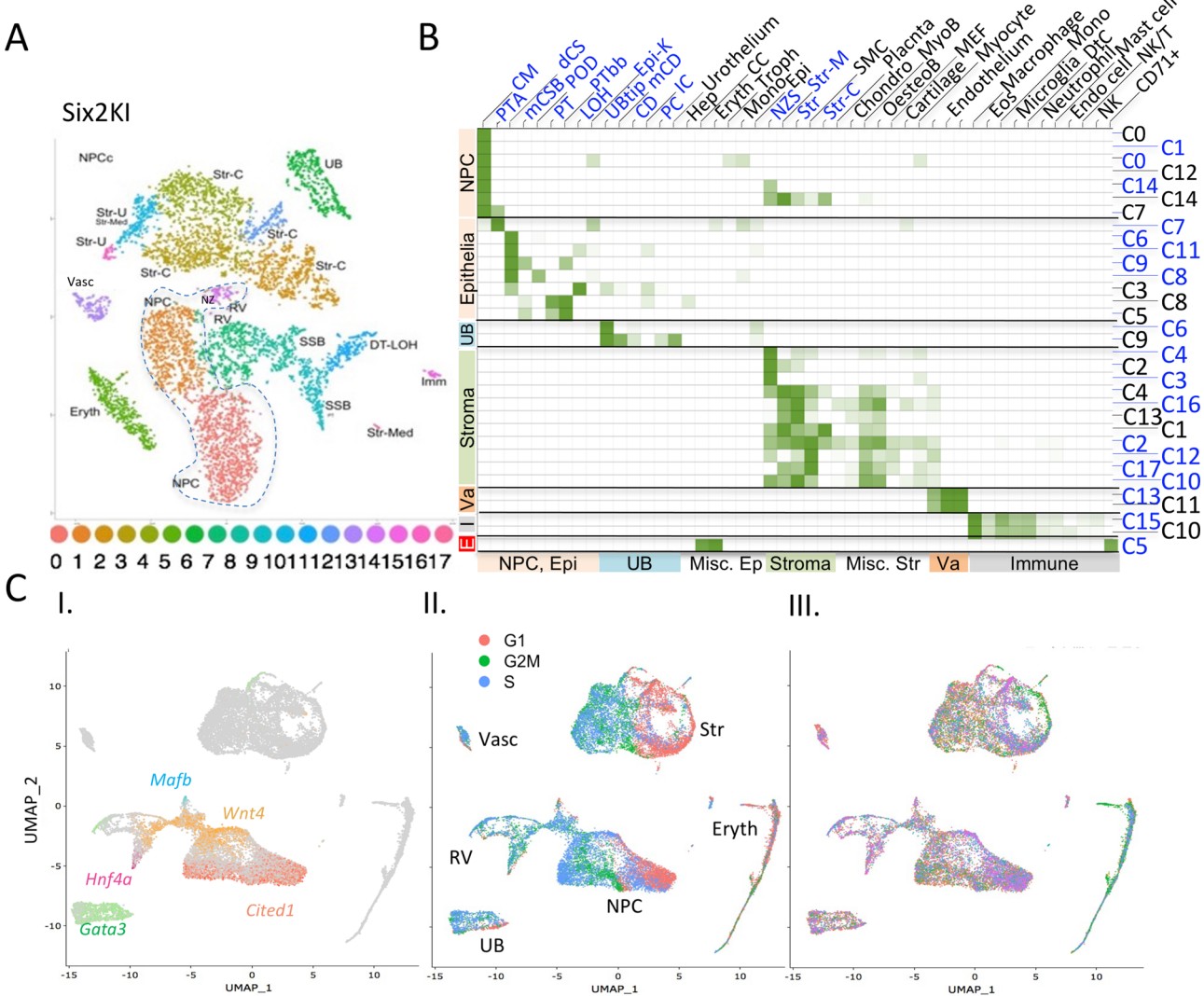

**Fig. 1 Cell populations identified in developing kidneys with different genetic makeup are indistinguishable. A** tSNE of E14 $Six2^{KI}$ data identifies 18 clusters. Annotation of clusters based on similarity to kidneys of the same genotype analyzed by Combes et al.[49] **B** GO-Elite analysis of Combes' (black, C0-C14) and our (blue, C0-C17) $Six2^{KI}$ clusters against global datasets. Note the cortical bias in our clusters. **C** UMAP clustering of all six samples. Gene expression patterns were merged in photoshop (i) to illustrate the temporal progression from naive ($Cited1+$, right) to epithelial ($Gata3+$, $Hnf4a+$, $Mafb+$) at left. Wnt4 marks transitional primed NPC. Cell-cycle states and cell type identities are shown in (ii), and the genotype contribution to each super-cluster in (iii). Note that contaminating erythrocytes are contributed predominantly by the green sample, E14 $Six2^{TGC;Tsc1}$ (color key in Fig. 2L).

inconsistent with the hypothesis that NPCs from $Six2^{KI}$ and $Six2^{TGC;Tsc1}$ retain a youthful, naïve character relative to control, and collectively suggest that the mechanism by which $Tsc1$ hemizygosity enhances nephron number does not involve enrichment in NPCs with a youthful transcriptomic profile.

We subsequently pooled all the NPC clusters identified based on their gene set enrichment analysis (GSEA) false discovery rate (FDR) scores (Supplementary Fig. S1B, red boxes). Each cell was uniquely identified and the combined NPC population re-clustered using all transcripts in Seurat 3.0. Six super clusters, each corresponding to a unique genotype/age combination, emerged. To ask if inclusion of all transcripts resulted in splitting of clusters, we applied six filtering permutations, excluding various combinations of mitochondrial, ribosomal, gene model, and/or blood transcripts (Fig. 2L). Regardless of the filter or clustering algorithm (tSNE or UMAP) applied, the same six super clusters were obtained. All P0 cells clustered together and separately from E14 cells, with $Six2^{KI}$ E14 segregating distinctly away from the other E14 samples. This analysis confirmed a key

finding underlying the tipping-point hypothesis:[38] that NPC transcriptomes are indeed altered over the 10 days of murine metanephrogenesis, including but not limited to the respiratory and translational machinery components. Furthermore, this analysis showed that the effects of $Six2$ and $Tsc1$ hemizygosity are profound enough to distinguish these cells not only from controls at both time points, but also from each other. This suggests that multiple pathways and distinct mechanisms may be responsible for the gene expression differences that separate these genotypes. Lastly, we note that while most ribosomal proteins (e.g., Rpl10a) were ubiquitously expressed irrespective of genotype or age, some ribosomal proteins exhibited intriguing distributions (Fig. 2M), which may not be captured when filtering out these gene types.

To ask if the mechanism by which $Tsc1$ hemizygosity in NPCs increased nephron number was reflected in the transcriptome, we evaluated gene expression signatures that might differentiate $Six2^{TGC;Tsc1}$ NPC clusters from the appropriate control, $Six2^{TGC}$ at postnatal day 0. Given the limitation of having a single pooled

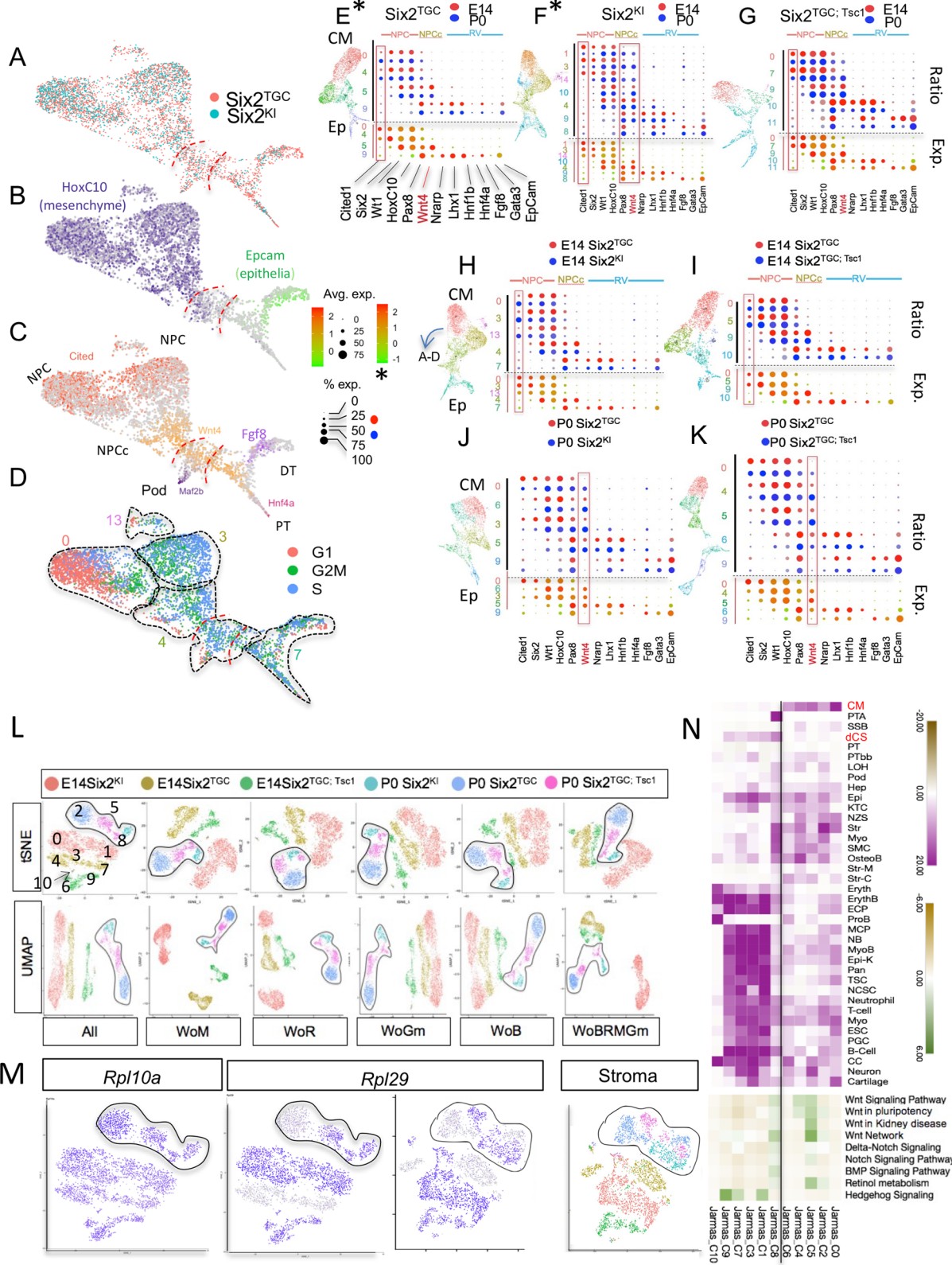

sample per condition, we split each sample into its male, female, and unassigned constituents to create pseudo-bulk replicates for each cell type (Supplementary Fig. S1H and Supplementary Data 6). Holistic differential expression analysis and visualization between all cell populations with the software cellHarmony[51] identified gene expression changes and pathway impacts consistent with the scRNA results from pooled embryos. The

top term distinguishing $Six2^{TGC;Tsc1}$ NPC from control at P0 was respiratory electron transport and heat production by uncoupling proteins (Supplementary Data 7 and Supplementary Fig. S1I). By this measure, $Six2^{TGC;Tsc1}$ NPC clusters appear older than $Six2^{TGC}$.

The emergence of ribosomal and metabolic signatures is consistent with the involvement of the mTOR signaling pathway.

**Fig. 2 Pair-wise UMAP integration analyses indicate that *Six2^{KI}* and *Six2^{TGC;Tsc1}* P0 NPC do not enrich for youthful character.** UMAP Integration analysis, similar to that shown in Fig. 1C, was performed on two samples at a time and the NPC super-cluster clipped from the image (integration of E14 *Six2^{KI}* and *Six2^{TGC}* is shown in **A–D**, **H**). **A** Distribution of NPC by genotype across the super-cluster at E14. **B** Expression of *Hoxc10* (mesenchymal, purple) and *Epcam* (epithelia, green) merged in photoshop illustrates MET across the cluster. **C** Gene expression patterns were merged in photoshop to illustrate the temporal progression from naïve (*Cited1*+, left) to epithelial (*Gata3*+, *Hnf4a*+, *Mafb*+, to the right). **D** Cell-cycle status of each NPC cluster (traced and numbered). Note that cluster numbers are represented on the Y axis of panels (**E–K**). **E–G** Dot plots for selected gene expression from pair-wise integration analysis of E14 and P0 NPC for each genotype. Violin plots are presented in Supplementary Fig. S2. On the left, colored numbers on the Y axis correspond to each NPC cluster, rotated to place naive on top and epithelial on the bottom. On the X axis, selected genes are arranged from left (naive) to right (epithelial). For each genotype, the fraction of cells expressing a given gene is reflected in the diameter of red (E14) or blue (P0) circles, ranging from 0% to 100%. The level of expression is plotted below in red (high) and green (low) circles, diameter range 0–75%, and scaled 0–2, or, where marked with an asterisk, from −1 to 2. Red box highlights gene expression. **H–K** Pair-wise integration analysis of E14 (**H**, **I**) or P0 (**J**, **K**) NPC for control *Six2^{TGC}* (red circles) vs. *Six2^{KI}* (blue circles, **H**; **J**) or vs. *Six2^{TGC;Tsc1}* (blue circles, **I**; **K**). Red box highlights gene expression. **L** Clustering all NPCs across six samples using tSNE and UMAP algorithms. All denotes no filter; WoM - without mitochondrial genes; WoR - without ribosomal genes; WoGm - without gene model transcripts; WoB - without blood transcripts; WoBRMGm - without all these entities. P0 clusters are traced. See text for details. **M** Rpl10a has a ubiquitous expression pattern across all NPC but note selective enrichment of Rpl29 in the high nephron number lines, *Six2^{TGC;Tsc1}* and *Six2^{KI}*, at both E14 and P0 in NPC and stroma. **N** GO-Elite analysis of NPC clusters for cell type (purple) or selected pathways (green).

We previously used a genetic test comparing nephron numbers in *Six2^{TGC;Tsc1}* and *Six2^{TGC;Tsc1,Mtor}* to propose independence of hamartin's effect on nephron endowment from mTorc1. However, the observation that the ribosomal protein *Rpl29* (Fig. 2M) was consistently elevated in NPC producing higher NN, as well as the stroma from these lines (scale favors display of the cells with expression above median) is consistent with differential translation in these cells which may reflect change in mTorc1 activity[52]. To evaluate mTOR signaling status in *Six2^{TGC;Tsc1}* NPC, we visualized mTOR activity indirectly via immunofluorescent staining for the phosphorylated ribosomal protein S6 (p-rpS6; Fig. 3A). As expected, complete loss of *Tsc1* in *Tsc1^{−/−}* NPCs (*Six2^{TGC}; Tsc1^{f/f}*) led to increased p-rpS6 staining in NPC, and removal of one *mTor* allele (*Six2^{TGC}; Tsc1^{f/f}; Mtor^{+/f}* NPC) restored p-rpS6 staining back to basal levels confirming mTorc1 activity was elevated and it is reduced by removal of one mTor allele in this genetic model. Notably, *Tsc1* hemizygote NPC do not exhibit detectable change in mTOR activity using this surrogate measure. Intriguingly, we detected evidence of elevated mTOR activity in stromal cells adjacent to compound hemizygote *Six2^{TGC;Tsc1,Mtor}* NPCs (compare Fig. 3A, A′ to Fig. 3C, C′), consistent with crosstalk between genetically manipulated NPC and wild-type stroma playing a role in the phenotypes we observed. Taken together, we propose that (1) a translational phenotype in *Tsc1* hemizygotes is not reflective of global increase in mTOR pathway activation and (2) the stromal compartment is responsive to genetic perturbations in the NPC.

Given the distinct clustering of E14 *Six2^{KI}* cells away from age-matched cells, we surmise that the increased nephron endowment in *Six2^{KI}* might be driven by a mechanism acting earlier in nephrogenesis, before P0. By contrast, our analysis suggests that the mechanism enhancing nephron number in *Six2^{TGC;Tsc1}* acts late, delaying cessation. To further identify global trends from our datasets, the top markers of each cluster were analyzed using the AltAnalyze GO-Elite function, which examines Gene Ontology overrepresentation covering gene, disease and phenotype ontologies, multiple pathway databases, biomarkers and transcription factors, and microRNA targets[53]. This analysis assigned to each NPC cluster, the cell types most highly correlated with its gene signature (Fig. 2N, purple, and Supplementary Data 8). Interestingly, this analysis resolved the NPC clusters into two distinct groups: A cap mesenchyme group (the E14 clusters C0, C4, C6, and the P0 clusters C2, C5), and an early epithelia group. All the P0 *Six2^{KI}* NPC, together with a few *Six2^{TGC;Tsc1}* cells, resembled PTA cells (cluster C8; cluster numbers shown in Fig. 2L). The rest of the NPC clusters (C1, C3, C7, C9, and C10) were classified as early epithelia, resembling distal comma-shaped body (Fig. 2N,

dCS), and enriched in transcripts contained in an eclectic collection of stem cells, epithelial cell types, and immune cells. Importantly, GO-Elite wiki-pathway analysis identified the highest enrichment for the WNT signaling (*Fhl2, Fzd2, Map1b, Tcf4, Wnt4*) and retinol metabolism pathways (*Aldh1a, Crabp1, Rbp1*) in *Six2^{TGC;Tsc1}* cells (cluster C5) compared to the other NPCs (Supplementary Data 8). The entire CM group was also enriched for ribosomal proteins, BMP, and Notch signaling, whereas the non-CM NPCs were enriched for ontologies related to Hedgehog signaling, DNA replication, and one carbon metabolism (Fig. 2N and Supplementary Data 8). Electron transport and oxidative phosphorylation pathways were enriched in non-CM E14 *Six2^{TGC}* clusters and P0 *Six2^{TGC;Tsc1}* CM, inconsistent with reduced respiration and driving increased nephron endowment in the latter population[43]. Indeed, we detected no differences in the metabolic activity of P0 control and *Six2^{TGC;Tsc1}* NPCs in Seahorse glycolysis and mitochondrial respiration assays (Fig. 3B, C and Supplementary Fig. S3). Although we note that the metabolic state of cultured NPCs is comparable to freshly isolated NPCs[43], the Seahorse assay may reflect adaptation to culture conditions such as oxygenation[54], particularly in the context of our specific allele (*Tsc1*). Thus, we opted to probe respiration status in vivo by an orthogonal method, pimonidazole injections. Pimonidazole binds to thiol-containing proteins in hypoxic environments[55] and can be used to identify differences in the degree cells are reliant on glycolysis vs. mitochondrial respiration. P0 control and *Tsc1^{+/−}* NPCs had indistinguishable hypoxic scores (Fig. 3E), determined by the pimonidazole-based fluorescence intensity in NPC niches relative to adjacent, hypoxic proximal tubules (Fig. 3D), inconsistent with the hypothesis that *Tsc1* hemizygote NPCs are more reliant on glycolysis than the controls thus delaying exit from the niche.

Collectively, these analyses did not uncover evidence to support the hypothesis that P0 *Six2^{TGC;Tsc1}* progenitors were more youthful than controls. Quite to the contrary, P0 *Six2^{TGC;Tsc1}* cells contained higher levels of Wnt4 mRNA, with signatures indicating elevated Wnt pathway activity (Fig. 2N) associated with MET and differentiation. However, they were still clearly CM cells (unlike P0 *Six2^{KI}* NPCs), with similar levels of mTorc1 activity and respiration to controls, raising the possibility that postnatal post-transcriptional events not captured by scRNA-Seq may be involved.

**Progressive translation bias in *Tsc1^{+/−}* increases Wnt signal.** No significant changes between E14 and P0 in the contribution of various signaling pathways involved in nephrogenesis including WNT (canonical and non-canonical), FGF, and GDNF was

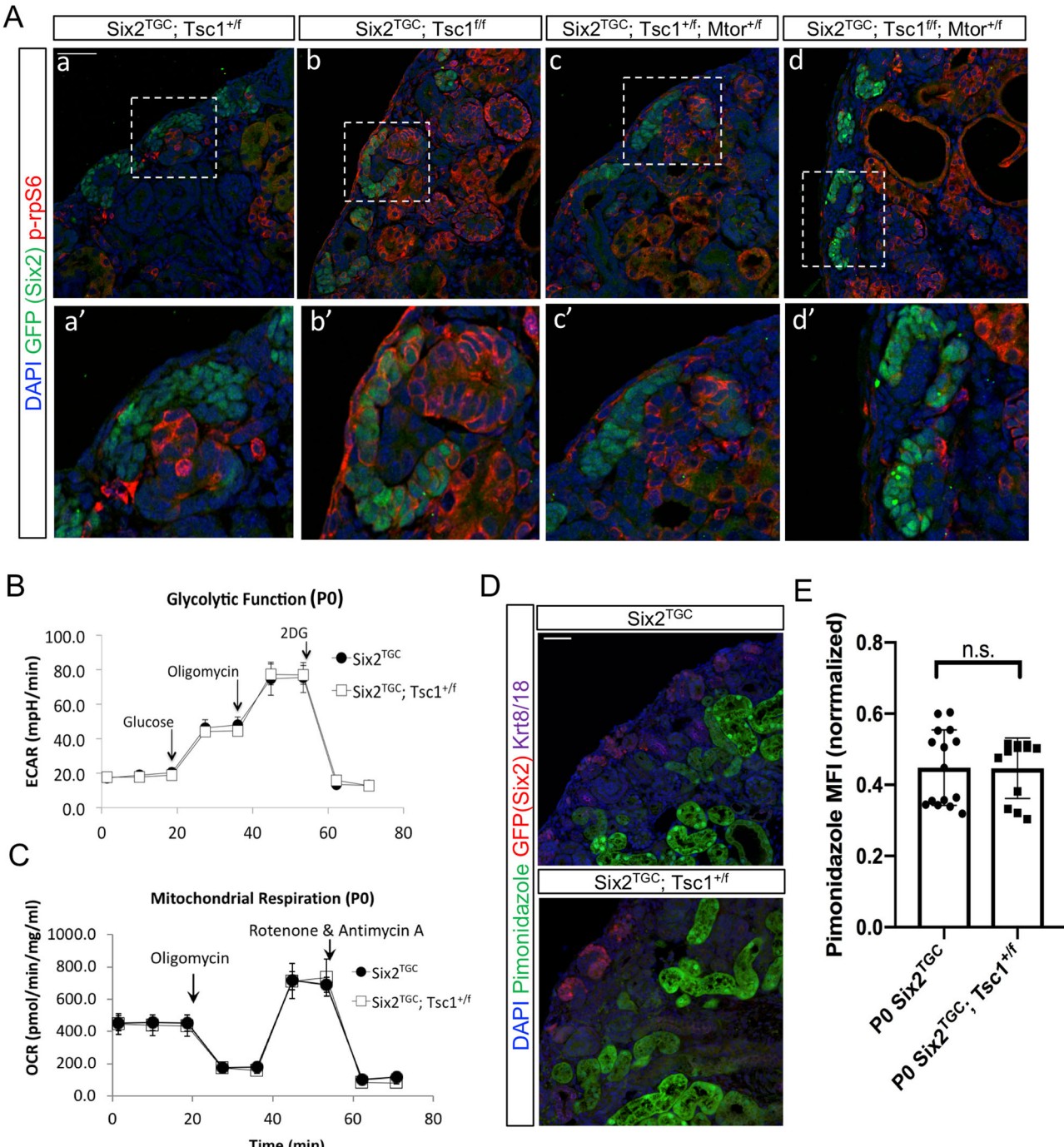

**Fig. 3 Evaluation of metabolic contributions to the enhanced nephrogenic phenotype in *Six2^TGC;Tsc1*. A** Immunofluorescence staining for the phosphorylated ribosomal protein S6 (p-rpS6, red) at P0 to assess mTorc1 activity in *Tsc1* hemi- and homozygous NPC. Genotype noted on the top, key to antibody staining on the left. Scale bar denotes 50 μm. Magnified view of NPC (green) in the dashed area below each panel. **B, C** Seahorse analysis of cultured NPCs reveals no change in glycolytic function ($n = 3$ control *Six2^TGC* and $n = 3$ *Six2^TGC;Tsc1* animals) or in mitochondrial respiration ($n = 3$ control *Six2^TGC* and $n = 4$ *Six2^TGC;Tsc1* animals); error bars represent standard deviation from sample group mean. **D** Representative image of staining by the hypoxia indicator pimonidazole reveals strong staining in the proximal tubules relative to the GFP (Six2) expressing NPCs. Scale bar denotes 50 μm. **E** Quantification of pimonidazole staining intensity in NPC normalized to adjacent tubular regions to control for differences in dose efficacy ($n = 4$ littermate animals in sample group; $n = 15$ control *Six2^TGC* and $n = 12$ *Six2^TGC;Tsc1* niches measured). Two-tailed unpaired *t* tests were performed in GraphPad Prism version 8 to evaluate statistical significance of pimonidazole signal; n.s. denotes not significant ($p = 0.9613$). Error bars represent standard deviation from the mean. Source data are provided as a Source Data file.

detected by CellChat[56], a package developed to probe signaling pathway activity in scRNA-Seq datasets (Supplementary Fig. S4A, B). Given the technical limitations of scRNA-Seq, including discordance with the protein state of the cell[57], and the suggestive

scRNA-Seq differentially expressed gene (DEG) signature pointing to a possible contribution of ribosomal constituents, we hypothesized that age-dependent changes in the signaling environment may be post-transcriptionally regulated. We thus shifted

to performing Translating Ribosome Affinity Purification (TRAP[58]) which has proven to provide a transcript profile with high concordance to the proteome. Having failed to detect NPC expression in Notch2-TRAP (obtained from http://www.gensat.org/) engineered to contain the ribosomal protein L10a EGFP fusion (EGFP/Rpl10a), we pursued a strategy utilizing mice containing EGFP/Rpl10a preceded by a floxed stop cassette in front of the powerful CAGGS promoter[59]. We noted some variability in ribosomal protein transcript composition between the genotypes, which prompted us to confirm that the ribosomal protein Rpl10a was indeed expressed in all cells of all genotypes and at both time points (Fig. 2M). The TRAP mice were crossed to $Six2^{TGC}$ or $Six2^{TGC;Tsc1}$ to activate expression of EGFP/Rpl10a in the NPC and their descendants. To analyze the translatome, cells were sorted for expression of GFP$^{HI}$ ($Six2^{TGC}$ and EGFP/Rpl10a, the latter also expressed in nephron epithelia) and Itga8$^{HI}$ (NPC, a few GFP- stromal cells), with negative selection for Pdgfra (stromal marker, Supplementary Fig. S4C). To overcome the potential confounding effects of isolation and sorting, cyclo-heximide was added to faithfully freeze elongating ribosomes and prevent their loss during the process. The ribosomes were pulled down from E14 control ($Six2^{TGC}$) and P0 animals of both genotypes ($Six2^{TGC}$ or $Six2^{TGC;Tsc1}$), allowing us to evaluate age-dependent changes, as well as capture effects driven by $Tsc1$ in later nephrogenesis, and the RNA was isolated as described[58].

RNA-Seq revealed significant differences between the E14 and P0 translatomes. For example, GO term analysis of differentially expressed genes identified a strong signature of cell cycle regulators in E14 $Six2^{TGC}$, and translation of signaling pathway components (GO:0009719, GO:0071495, GO:0007167, $p < 10^{-14}$, FDR $< 10^{-10}$) that were significantly enriched in P0 $Six2^{TGC}$ (Fig. 4A, Supplementary Fig. S4D, and Supplementary Data 9), a trend becoming more pronounced in P0 $Six2^{TGC;Tsc1}$. Most GO terms specifically enriched in P0 $Six2^{TGC;Tsc1}$ were related to neurogenesis, due to the prominence of genes associated with axon pathfinding (Supplementary Fig. S4D, E and Supplementary Data 9).

Consistent with the tipping-point hypothesis, and with age-dependent post-transcriptional regulation, translation of components in all signaling pathways was elevated in older P0 $Six2^{TGC}$ NPCs (Supplementary Fig. S4D). Notably, most, but not all transcripts were recovered by TRAP more robustly in P0 $Six2^{TGC;Tsc1}$ relative to P0 $Six2^{TGC}$ NPCs, again consistent with the finding that these progenitors are closer to an epithelial transition (elevated Hippo and Notch signaling) and less youthful overall. Among the more abundant transcripts in $Six2^{TGC}$ at P0 were many Wnt agonists (green boxes, green dot marks non-canonical agonist). By contrast, in P0 $Six2^{TGC;Tsc1}$ transcripts of Wnt antagonists (red dots) were significantly more abundant in the translatome. They included $Dkk1$, $Kremen1$, and $Kremen2$, which together target the Lrp co-receptors for degradation (arrows in Fig. 4B), and $Znrf3/Rnf43$ which, in the absence of R-spondin/Lgr5 proteins targets the Wnt/Fzd complex for degradation (all marked with red dots in Fig. 4B[60]). Importantly, we noted that only $Rspo3$, $Tmem59$, $Lrp5$, $Wnt7b$, $Plaur$, and $Sf3b5$ were altered in the translatome in P0 $Six2^{TGC;Tsc1}$ relative to P0 $Six2^{TGC}$ NPCs by more than 1.2-fold, and had an adjusted $p$-value < 0.05 (as shown in Supplementary Data 9). $Rspo3$ and $Tmem59$ are key Wnt agonists (blue box, Fig. 4B): $Rspo3$ is required to engage $Znrf3/Rnf43$ and prolong the half-life of the Wnt/Fzd complex, and $Tmem59$ encodes a molecule thought to cluster Fzd proteins thus increasing avidity/signal strength of WNT signaling[42]. The combination of increased antagonists and decreased agonists was unique to the Wnt signal transduction pathway (Supplementary Fig. S4C, D). Combined, these observations suggest the hypothesis that hamartin/$Tsc1$ regulates some aspects of the Wnt9b signal by affecting the translation of key transcripts, and the impact on nephrogenesis in $Tsc1$ hemizygotes might be due to altered perception of Wnt signal input to below the threshold required for completion of MET, delaying NPC exit from the niche as well as cessation of nephrogenesis.

To test the hypothesis that Wnt signaling components differentially represented in the translatome of $Tsc1$ hemizygote NPCs underlie the mechanism involved in nephron gain, we first asked if increased transcript abundances in the translatome corresponded to increased protein abundance (Lgr5 and Fzd10). Flow cytometry of dissociated, fixed, and permeabilized control NPCs confirmed the concordance of the intracellular pool with the translatome (Fig. 4B'). Note that Lgr5, Fzd10, and Rspo3 mRNA levels were not different by RT-qPCR on mRNA isolated from sorted NPCs, consistent with a post-transcriptional mechanism (Fig. 4C). Given the observed reduction of $Rspo3$ in the translatome, we predicted that mice with hemizygote NPC, anticipated to decrease Wnt/Fzd complex stability, would gain nephron numbers (note that $Rspo3$ nulls exhibit MET impairment[61]). We tested this prediction in $Six2^{TGC}; Rspo3^{+/f}$ by counting nephrons using the acid maceration method[62] in young adult (≥P28, Supplementary Fig. S1D) mice. As shown in Fig. 4F, removal of one $Rspo3$ allele in NPC resulted in an increase in nephron number similar in magnitude to that seen in $Six2^{TGC;Tsc1}$ mice. $Rspo3$ is also produced in the stroma; as an additional control, we examined $FoxD1^{Cre}; Rspo3^{+/f}$, to ask whether stromal Rspo3 affected nephron numbers to any degree. We observed no change in nephron numbers when the $Rspo3$ allele was removed from the stroma, indicating that this factor, like Fgf20, acts in a juxtacrine manner (Fig. 4F). Furthermore, GFP+ NPCs are scarcely observed at P4 in $Six2^{TGC}$ controls[40], but GFP+ NPC persist in kidneys from $Rspo3$ hemizygote littermate (Fig. 4D and Supplementary Fig. S4F), just as we reported occurred in $Six2^{TGC;Tsc1}$ mice[40]. Thus, reduction in $Rspo3$ levels recapitulates both the increase in nephron numbers and the delayed cessation phenotype observed in $Tsc1$ hemizygotes.

If reduced translation of $Tmem59$ contributed to the phenotype observed in $Six2^{TGC;Tsc1}$, we would predict heterozygosity for $Tmem59$ would enhance nephron numbers alone or synergistically with $Rspo3$. Since generation of $Tmem59$ ($Dcf1$) knockout mice was reported to produce viable animals[63], we designed sgRNAs to disrupt $Tmem59$ Exon4 (shared by all alternative transcripts) to create a deletion and/or missense mutations. The gRNA oligos and dCas9 were co-injected into fertilized CD1 eggs. From the 18 CRISPR F0 offspring, genotyped by cloning PCR products from F2 offspring and sequencing (Supplementary Fig. S5A, B), we selected three alleles for analysis (Supplementary Data 10); 13 bp deletion (13bpDEL, missense), and 69 bp deletion (69bpDEL) and 822 bp deletion (822bpDEL) both of which result in an in-frame exon-skipping. We analyzed mRNA levels by RT-qPCR and confirmed the presence of the mutation in the transcript by sequencing cDNA (Supplementary Fig. S5C). The 13bpDEL homozygotes did not contain mRNA molecules encompassing exons 7-8, consistent with premature termination (Supplementary Fig. S5D). However, three commercial antibodies failed to detect the predicted molecular weight or the absence/truncation of protein in homozygous 13bpDEL animals, precluding character-ization of mutant protein levels (Supplementary Fig. S5E). While both $Tmem59^{13bpDel/13bpDel}$ and $Tmem59^{822bpDel/822bpDel}$ display a significant increase in nephron numbers (Supplementary Fig. S5F, G), consistent with the proposed agonist role of Tmem59 and the impact of reduced Wnt signal on cessation and indicative of a negative role in determining nephron numbers, $Tmem59^{+/13bpDel}$

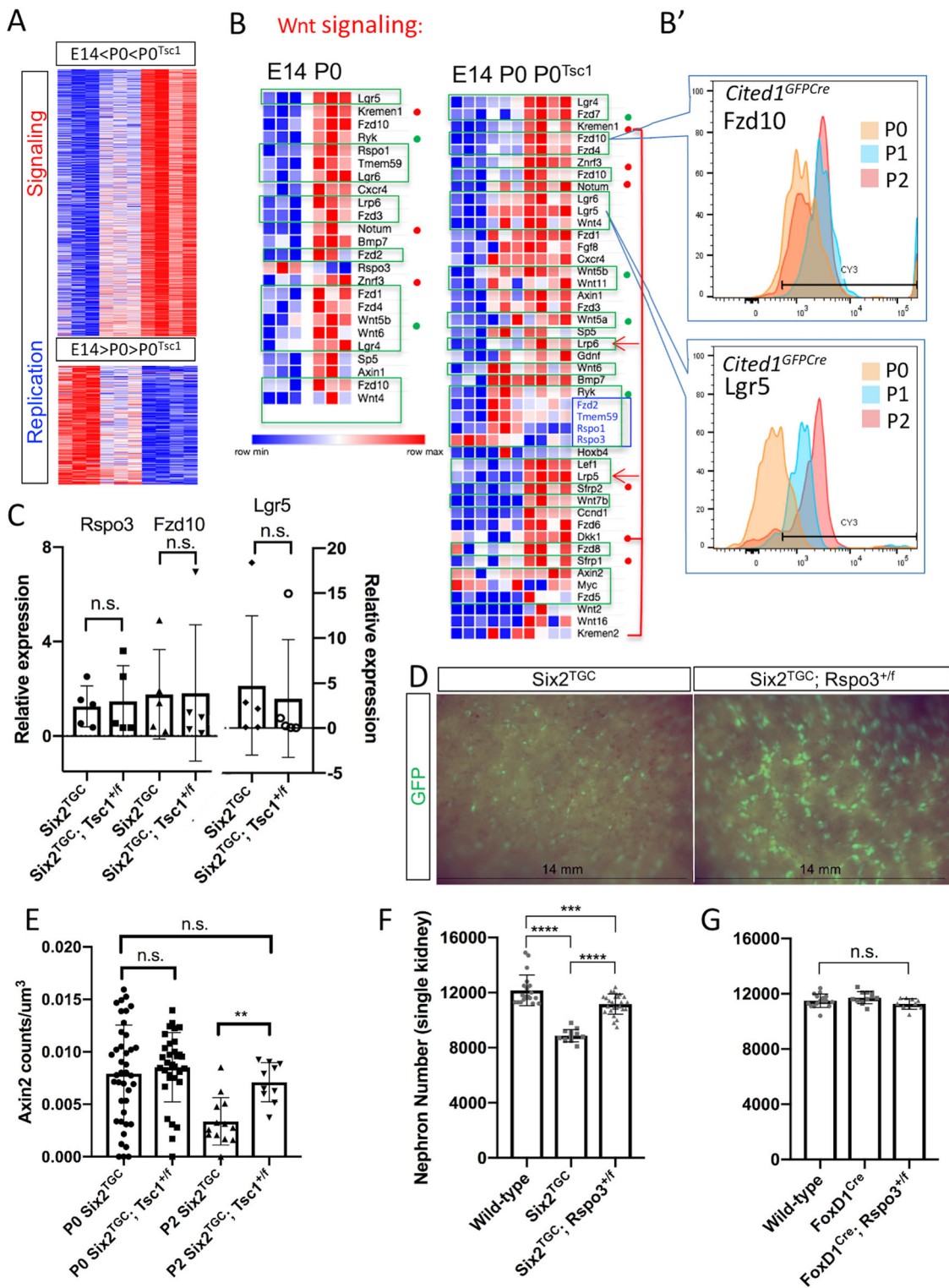

and *Tmem59*$^{+/822bpDel}$ kidneys did not gain nephrons compared to controls. Importantly, compound *Six2*$^{TGC}$; *Tmem59*$^{+/822bpDel}$; *Rspo3*$^{+/f}$ kidneys did not display synergy in regulation of nephron numbers (Supplementary Fig. S5H).

Despite the fact that young and old NPCs display different self-renewal/replication rates[37], classical readouts of Wnt activation (like Axin2) may not reflect this regulation accurately. Although some known Wnt targets were generally upregulated in P0 *Six2*$^{TGC;Tsc1}$ NPC, quantification of Axin2 transcripts by RNA in situ hybridization (RNAscope) in NPCs at P0 and P2 revealed that while Axin2 levels decreased with age in control *Six2*$^{TGC}$, Axin2 levels in P2 *Six2*$^{TGC;Tsc1}$ resembled the younger cells (Fig. 4E). Accordingly, Axin2 was not differentially bound to RPL10a in *Six2*$^{TGC}$ E14 vs. P0 NPC. We propose that relevant Wnt targets associated with balancing self-renewal and differentiation may be present among the transcripts inversely translated in P0 *Six2*$^{TGC;Tsc1}$ relative to P0 *Six2*$^{TGC}$ (Supplementary Data 9).

**Fig. 4 Translatome analysis and genetic validation models identify Wnt signaling as a likely mediator of enhanced nephrogenesis in *Six2^{TGC;Tsc1}***
**animals. A** Differentially translated transcripts in sorted NPC isolated from E14 *Six2^{TGC}*, P0 *Six2^{TGC}*, and P0 *Six2^{TGC;Tsc1}*. The data are shown as relative change by row, with full expression data shown in Supplementary Data 9. Top representative GO term for subset of genes is shown to the left, see Supplementary Fig. S4 for breakdown of ten signaling pathways. **B** On the left, a comparison of differentially translated Wnt pathway transcripts between E14 and P0 *Six2^{TGC}* highlights gains in signal-reception machinery (agonists in green boxes, antagonists marked with red dots, non-canonical signaling marked with green dots). On the right, comparing all three samples. Note the increase in antagonists (beyond the change observed between E14 and P0 control cells) and the decrease in agonists in P0 *Six2^{TGC;Tsc1}*. **B'** Specific proteins selected for validation by flow cytometry to detect changes in NPC levels between P0, P1, and P2, with good concordance to patterns observed in the translatome data. **C** RT-qPCR for selected transcripts on RNA isolated from P0 NPCs reveals no expression differences in *Rspo3*, *Fzd10* and *Lgr5*. Two-tailed unpaired *t* tests were performed in GraphPad Prism version 8 to evaluate statistical significance; n.s. denotes not significant (for *Rspo3*, $p = 0.7888$; for *Fzd10*, $p = 0.9701$; for *Lgr5*, $p = 0.7584$). $n = 5$ independent biological samples per group, with each sample representing RNA isolated from cells pooled from 3 littermates of the same genotype. Error bars represent standard deviation from the mean. **D** Surface images of GFP+ nephrogenic niches of late P4 kidneys from littermates (see also Supplementary Fig. S4F) detects persistent GFP indicating delayed cessation in *Rspo3* hemizygote kidneys. **E** *Axin2* transcript counts by RNAscope normalized per volume of Six2+ NPC niches at P0 and P2 reveals no change between genotypes at P0, with P2 *Tsc1^{+/−}* niches resembling the younger (P0) controls. At each time point, $n = 3$ independent animals were included per genotype. One-way ANOVA with Tukey's multiple comparisons, and with Welch's correction tests were performed in GraphPad Prism version 8 to evaluate statistical significance; n.s. denotes not significant, $^{**}p \leq 0.01$. At P0, $p = 0.9089$; at P2, $p = 0.0015$; and for the P0 *Six2^{TGC}* and P2 *Six2^{TGC;Tsc1}* comparison, $p = 0.8353$. Error bars represent standard deviation from the mean. **F, G** To validate the prediction that nephron number is dependent on a Wnt9b signal threshold, which progressively rises as translation of key components is increasing, we analyzed nephron numbers in adult (≥P28) mice with (**F**) NPC hemizygous for *Rspo3* ($n = 10$ wild type, $n = 6$ *Six2^{TGC}* and $n = 13$ *Six2^{TGC}; Rspo3^{+/f}* animals); for wild-type vs. *Six2^{TGC}; Rspo3^{+/f}* $p = 0.0006$, else $p \leq 0.0001$ or (**G**) stroma hemizygous to *Rspo3* ($n = 8$ wild type, $n = 6$ *FoxD1^{Cre}* and $n = 5$ *FoxD1^{Cre}; Rspo3^{+/f}* animals), $p = 0.4076$. Error bars represent standard deviation from the mean parameter values. One-way ANOVA with Tukey's multiple comparisons tests were performed in GraphPad Prism version 8 to evaluate statistical significance of single-kidney nephron numbers; n.s. denotes not significant, $^{***}p \leq 0.001$, $^{****}p \leq 0.0001$. Source data are provided as a Source Data file.

## Discussion

The development of therapeutic interventions aiming to prolong nephrogenesis in premature birth will require a detailed molecular understanding of nephron progenitor niche retention and exhaustion. Transcriptome analysis at the single-cell level is a powerful tool applied effectively to build detailed maps of cellular constituents and their developmental trajectories in various tissues. While multiple studies have generated single-cell transcriptomic data for the developing mouse[44,49,64,65], rhesus[66], and human[67,68] kidney, the dominant focus in these early phases has been on enumerating the cell types comprising the kidney, with mechanistic exploration delegated for future investigation.

Collectively, TRAP, scRNA-Seq of samples from two developmental time points across three genotypes, and in vivo genetic validation led to several important conclusions. First, we observed that nephron endowment can be influenced by either molecular changes occurring early in nephrogenesis (as in *Six2* hemizygotes) or late in nephrogenesis (as in *Tsc1* hemizygotes), likely operating via distinct mechanisms. Second, a plausible mechanism impacting the timing of cessation is age-dependent increased translation of receptors, co-receptors, ligands, and other signal-reception-enhancing proteins across all the ten pathways we examined (Notch, Wnt, FGF, BMP, TGF, HH, YAP, RA, PDGFR, VEGF). Note that, when tested, the transcripts of Wnt signaling components were not significantly different between young and old NPC. Thus, while age may change the niche signaling milieu —for example, less stromal *Dcn*—a key component in the tipping point appears to be increased awareness in older NPCs of their existing signaling environment. These translational changes may work synergistically or independently from the Lin28/Let7 axis[69] or metabolism[43,70]. Third, reduction in *Tsc1* dose correlated with enhanced translation across all signaling pathways, and NPC appeared older and more PTA-like than control. Fourth, unexpectedly, *Tsc1* hemizygosity also resulted in elevated *Rpl29* expression and in asymmetric association with the Rpl10 translational machinery of several transcripts, including Wnt singling components, which may lead to key agonists (R-spondin proteins) being translated less efficiently and key antagonists (Dkk1/Kremen, Znrf3/Rnf43) translated more efficiently. Intriguingly, the regulation of translated transcripts by reduced *Tsc1* dosage is

not associated with global elevation in mTOR activity within NPCs. How Rpl29 and hamartin impact this asymmetry—and whether such a phenomenon is observed in other developmental or cellular contexts—are topics of active investigation. Fifth, reduced *Rspo3* dosage recapitulated the *Tsc1* hemizygous phenotype (delayed cessation and increased nephron number), consistent with reduced translation of Rspo3/1 in *Tsc1* mutant NPC. Sixth, the genetically intact stroma is responding to changes in the genotype of the adjacent NPC consistent with crosstalk between the two tissues[28,29].

It has been long-appreciated that a UB-derived Wnt9b canonical signal is required for both maintenance of self-renewal capacity of NPC, as well as for their differentiation[71,72]. Recently, work in cells derived from E10.5 mesonephric mesenchyme identified a role for Rspo1, Lrp6, and Fzd5 in enabling Wnt9b-responsiveness as cells transitioned from the intermediate mesoderm to the metanephric mesenchyme identity[73]. It was further demonstrated that low signaling input promotes self-renewal and high signaling input promotes differentiation[74]. Our findings explain how Wnt9b signal strength varies to control both decisions: a cell-intrinsic and progressive increase in translation leads to increased stability of Wnt/FZD complexes via elevated Rspo, Lrp, and Lgr proteins. This enables a cell to receive a more sustained and/or stronger Wnt9b signal. Coupled with additional changes yet to be characterized, the entire population switches from self-renewal to differentiation. We validated this by demonstrating that reduced *Rspo3* recapitulated the *Tsc1* phenotype. However, how can we reconcile this cell-intrinsic process with the observation that old cells in a young neighborhood continue to self-renew[38]?

Since modulation of *Rspo3* dosage within the FoxD1 lineage did not impact nephron number but *Rspo3* hemizygosity in NPC did, we speculate that one mechanism explaining how young cells enable old cells to remain in the niche[38] is by providing juxtacrine Fgf20 (in addition to Fgf8/9[75]) and diluting Rspo1/3 below the threshold required to send NPC to exit. Note that Fgf8/9 production is elevated at P0, perhaps because Fgf20 alone cannot counter the increase in Wnt9b signal received. Together, the data suggests that a very precise threshold needs to be met before cells lose their ability to self-renew and fully commit to differentiation.

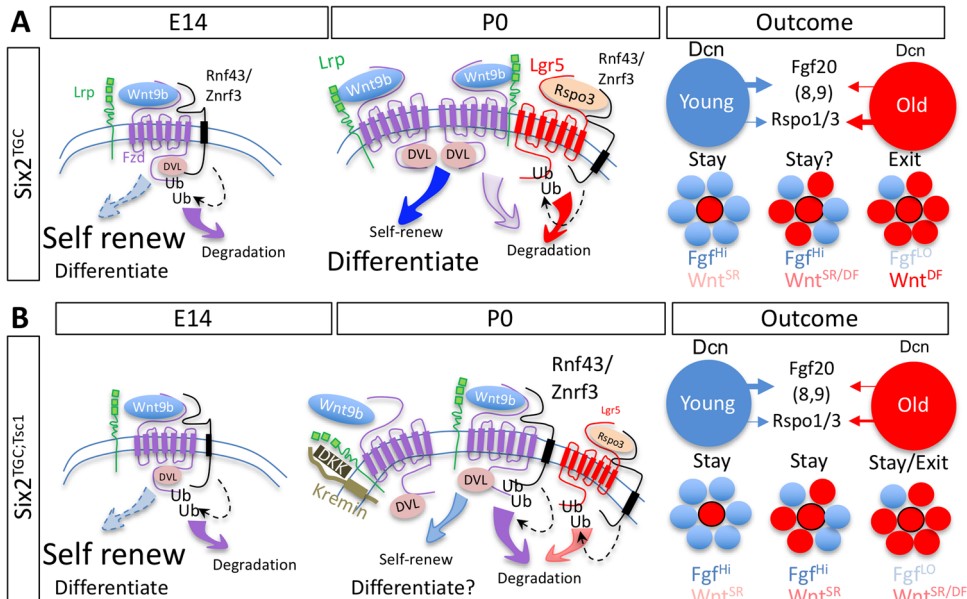

**Fig. 5 A parsimonious hypothesis of how NPC coordinate the cessation of nephrogenesis. A** In E14 *Six2*^TGC controls, important Wnt agonists are transcribed but poorly translated, resulting in lower Wnt signal intensity and preference for self-renewal (SR). Transcript levels for the niche-retention signal Fgf20 are relatively high[38]. At P0, the increased translation of Rspo1/3 and additional factors increase the stability of Fzd/Wnt complexes; Wnt signal intensity now promotes differentiation (DF) over self-renewal. We assume other factors in the niche environment, including Decorin, can modulate this in trans. If enough neighbors are in this state, an environment low in FGF and high in Rspo1/3 will drive the NPC to differentiate en masse. **B** In *Six2*^TGC;Tsc1 cells, translation of Rspo1/3 is less efficient, and that of antagonists DKK, Kremen, and RNF43 is more efficient. As a result, cells in the population are experiencing lower Wnt signals, delaying coordinated cessation by an unknown amount sufficient to produce more nephrons. Experimental data suggest this delay is ~12 h. This environment may also increase the number of nephrons forming during arcading (the period during which the entire population remaining in the niche differentiates[32]).

The outcome of Wnt9b reception is inherently dependent on the combined expression patterns of multiple proteins, including secreted agonists/co-factors provided in trans, coupling all the NPC within the niche (Fig. 5).

The implication of this study to human health is clear: it demonstrates that a subtle reduction in Wnt signaling can increase nephron endowment, expanding the possible therapeutic approaches to address clinical need in treatment of premature infants in addition to minimizing nephrotoxic insults. A similar observation was reported with a BMP antagonist, Decorin[29]. Given the centrality of Wnt and BMP to ongoing neurological developmental processes, these may not be the ideal target pathways, though this remains to be explored. Importantly, *Six2* hemizygote NPC are transcriptionally distinct from *Tsc1* hemizygote NPC, suggesting that additional pathways operating at different time points all contribute to regulating nephron endowment, only some through extension of cessation timing or enhancement of late-stage nephrogenesis. Of note, a later-acting mechanism may reflect a more targetable period of nephrogenesis for therapy in premature infants as more organ systems complete their development. Future investigation into the mechanism by which hamartin selectively alters translation may identify specific, safer targets for therapeutic intervention to elevate nephron endowment.

## Methods

### Animal studies

*Mouse strains and embryo staging.* Our experimental protocols (IACUC2016-0022/0032 and IACUC2018-0107/0108) are approved by the Institutional Animal Care and Use Committee of Cincinnati Children's Hospital Medical Center (CCHMC) and all experiments herein comply with ethical regulations for animal testing. The following lines were used in this study: *Tg(Six2-EGFP/cre)1Amc* (Jax Stock No: 009600; herein *Six2*^TGC), Six2-CreERT2 (herein *Six2*^KI)[19], *Tsc1*^f/f;[76], B6;129S4-Gt(ROSA)26Sortm9(EGFP/Rpl10a)Amc/J (Jax Stock No: 024750; herein EGFP/

Rpl10a), *Mtor*^f/f;[77], *Rspo3*^f/f (Jax Stock No: 027313; Rspo3^tm1.1Jcob), *Cited1-CreERT2-GFP*^+/tg;[20], and *Foxd1*^tm1(GFP/cre)Amc (herein *Foxd1*^Cre; Jax Stock No: 012463). Isolated kidneys from Collaborative Cross (CC) strains (CC003, CC006, CC007, CC008, CC013, CC021, CC026, CC045, CC051, and CC071) and selected breeders were purchased from UNC Department of Genetics. All CC mice were obtained from the Systems Genetics Core Facility at the University of North Carolina (UNC)[78]. Prior to their relocation to UNC, CC strains were generated and bred at Tel Aviv University in Israel[79], Geniad LLC in Australia[80], and Oak Ridge National Laboratory in the United States[81]. CC mice can be obtained from the Systems Genetics Core Facility at UNC (http://csbio.unc.edu/CCstatus/index.py). In mouse experiments using embryos, embryonic day (E) 0.5 was designated as noon on the day a mating plug was observed. All mice were maintained in the CCHMC animal facility according to the animal care regulations. Animals were housed in a controlled environment with ambient temperature (22 °C) and humidity (30–70%), a 14 h light/10 h dark cycle, and free access to water and a standard chow diet. Genotyping was performed on DNA isolated from the tail, spleen, or liver using the primer sets reported in Supplementary Table 1.

*Generation of TMEM59 mutant mice.* The methods for the design of guide RNA and the production of animals were described previously[82]. We targeted *Tmem59* with a guide RNA (target sequence: ACGACCTCACCAGAGTCAGA) according to the on- and off-target scores from the web tool CRISPOR (http://crispor.tefor.net[83]). To form guide RNA duplex, 5 µl crRNA (500 ng/µl; Integrated DNA Technologies (IDT), Coralville, IA) and 10 µl tracrRNA (500 ng/µl; IDT) were mixed in the IDTE buffer and heated to 95 °C for 2 min, followed by slow cooling to 25 °C. To form ribonucleoprotein complexes (RNPs), 3 µl chimeric guide RNA and 3 µl Cas9 protein (1000 ng/µl; IDT) were added to 9 µl Opti-MEM and incubated at 37 °C for 15 min. The zygotes from super-ovulated female mice on the CD-1 genetic background were electroporated with 7.5 µl RNPs on ice using a Genome Editor electroporator (BEX Co. Ltd, Tokyo, Japan); 30 V, 1 ms width, and 5 pulses with 1 s interval. Two minutes after electroporation, zygotes were moved into 500 µl cold M2 medium (Sigma-Aldrich, St Louis, MO, USA), warmed to room temperature, and then transferred into the oviductal ampulla of pseudo-pregnant CD-1 females. 18 F0 CRISPR pups were born and genotyped by PCR and Sanger sequencing. Primers for genotyping are reported in Supplementary Table 1.

*Single-cell sample preparation and sequencing.* E14 and P0 kidneys were isolated in ice-cold PBS. The capsule was removed, four (P0) or six (E14) kidneys were placed in 1.5 ml Eppendorf tubes, digested using 250 µl of 10 mg/ml *Bacillus licheniformis* cold-protease (Sigma-Aldrich, St Louis, MO, USA) and incubated at 10 °C in an

Eppendorf Thermomixer R shaking at 1400 rpm. After 5–7 min, the kidneys were removed, and the single-cell suspension was rinsed twice with 750 µl of ice-cold 0.04% BSA/PBS. Trypan Blue stained cells were counted using a hemocytometer, requiring greater than 85% cell viability. Cells were resuspended at a concentration of 1000 cells/µl and loaded on a chromium 10x Single Cell Chip (10x Genomics, Pleasanton, CA, USA). Libraries were prepared using Chromium Single Cell Library kit V2 (10x Genomics) and sequenced on an Illumina HiSeq-2500 using 75 bp paired-end sequencing.

*NPC isolation for TRAP.* To perform ribosome affinity purification on NPCs, timed-matings between $Six2^{TGC}$ or $Six2^{TGC;Tsc1}$ and EGFP/Rpl10a mice were set, and embryos harvested at E14 or P0. All subsequent procedures were performed with solutions containing 100 µg/ml cycloheximide. Kidneys were isolated in ice-cold PBS and the capsule removed. Approximately 4–6 kidneys were placed in 1.5 ml Eppendorf tubes and digested with 250 µl of 0.25% Trypsin/0.2% Collagenase (Sigma-Aldrich) at 20 °C in a Thermomixer R shaking at 1400 rpm. After 10–15 min, the kidneys were removed, and the single-cell suspension was rinsed twice with 750 µl of ice-cold 0.3% BSA/PBS. NPCs were labeled using cell-surface primary antibodies (1:250 biotin-labeled anti-ITGA8, R&D Systems, Minneapolis, MN, Catalog #BAF4076; 1:50 PE-conjugated anti-PDGFRA, Invitrogen, Carlsbad, CA, USA; Catalog # 12-1401-81) while rocking on ice for 30–60 min. Cells were rinsed twice in 750 µl of ice-cold 0.3% BSA/PBS and subsequently stained with Streptavidin APC (1:50, Invitrogen; Catalog #SA1005) while rocking on ice for 30–60 min. After two washes in 750 µl of ice-cold 0.3% BSA/PBS, the NPCs were isolated by FACS on a SH800S cell sorter (Sony Biotechnology Inc., San Jose, CA, USA) selecting for GFP^HI (Six2 and EGFP/Rpl10a-expressing NPC), ITGA8+ (NPC with some stroma labeling, but not nephron epithelia, which remain EGFP/Rpl10a+), and PDGFRA- (Stroma is PDGFRA+) cells. To minimize cell stress during sorting, selection of an appropriately wide flow cytometer nozzle (100 µm), and precoating of collection tubes with collection buffer solution (ice-cold 0.3% BSA/PBS) were applied. Purification of polysome-bound RNA from kidneys was performed as previously described[84]. Briefly, 250,000–300,000 NPCs were homogenized in ice-cold lysis buffer (20 mM HEPES, 5 mM MgCl2, 150 mM KCl, 0.5 mM DTT, 1% NP-40), 100 µg/ml cycloheximide, RNase inhibitors (40 U/ml RNasin (Promega, Madison, WI, USA), 1 U/µl Recombinant RNase inhibitor (Takara Biosciences, Mountain View, CA, USA)), and protease inhibitors (1 tablet per 10 ml lysis buffer, complete mini EDTA-free protease inhibitor cocktail, Sigma) by vigorous pipetting. The samples were centrifuged at 2000g for 10 min at 4 °C to remove large debris. Supernatants were extracted with 1% NP-40, and 30 mM 1,2-diheptanoyl-sn-glycero-3-phosphocholine (DHPC, Avanti Polar Lipids, Alabaster, AL, USA) on ice for 5 min, and centrifuged at 20,000g. The clarified supernatant was incubated in low-salt buffer (20 mM HEPES, 150 mM KCl, 5 mM MgCl2, 0.5 mM DTT, 1% NP-40), RNAse inhibitors (40 U/ml RNasin (Promega) and 1 U/µl Recombinant RNase inhibitor (Takara Biosciences)), and 100 µg/ml cycloheximide containing streptavidin/protein L-coated Dynabeads (ThermoFisher, Waltham, MA, USA) bound with 17 µg of anti-GFP antibodies (AB_2716736, Catalog #HtzGFP-19F7 and AB_2716737, Catalog #HtzGFP-19C8, Memorial Sloan Kettering Centre, New York, NY, USA) overnight at 4 °C with gentle agitation. After 24 h, the beads were collected using magnets for 1 min on ice, washed four times in high-salt washing buffer (20 mM HEPES, 350 mM KCl, 5 mM MgCl2, 0.5 mM DTT, 1% NP-40, 40 U/ml RNasin (Promega) and 1 U/µl Recombinant RNase inhibitor (Takara Biosciences), and 100 µg/ml cycloheximide) and collected with magnets. RNA was eluted from the samples and purified using a Qiagen (Hilden, Germany) micro-RNeasy RNA kit per manufacturer's instructions. RNA integrity was assessed using an RNA PicoChip (Agilent Bioanalyzer, Santa Clara, CA, USA) and the Clontech SMARTer library kit V2 (Takara Bio, USA) was used for cDNA library construction. Libraries were sequenced on an Illumina HiSeq-2500 using 75 bp paired-end sequencing.

*Total protein quantification by flow cytometry.* Briefly, kidneys from P0, P1, or P2 *Cited1-CreERT2-GFP*^+/tg mice were isolated in ice-cold PBS and the capsule removed. Four to six kidneys were digested with 1 mg/ml neutral-protease and incubated at 20 °C for ~5–10 min in a Thermomixer R shaking at 1400RPM. The kidneys were examined under a fluorescent microscope to ensure removal of nephron progenitors (GFP+ cells). It is important to note that isolating NPCs for flow cytometry using other enzymatic means, such as cold-protease, trypsin, or collagenase had variable results. After digestion, the cells were washed twice with ice-cold 1% BSA/PBS, fixed with 4% PFA for 10 min, and washed twice with ice-cold 1% BSA/0.1% Triton-100/PBS. Permeabilized cells were incubated with the primary antibodies anti-Lgr5 (1:250, R&D Systems, Catalog #MAB82041), or anti-Fzd10 (1:250, ProteinTech, Catalog #18175-1-AP) and anti-GFP (1:500 Aves Labs, Davis, CA, Catalog # GFP-1020) on ice for 60 min. After incubation, the cells were washed twice with ice-cold 1% BSA/0.1% Triton-100/PBS and stained with secondary antibodies (1:500 anti-Rabbit Cy3, Jackson Immuno Research, West Grove, PA, USA, Catalog #711-165-152; and 1:500, anti-Chicken FITC, Jackson Immuno Research, Catalog #703-095-155) for 30–60 min while rocking on ice. The cells were washed twice with ice-cold 1% BSA/0.1% Triton-100/PBS and the cells were analyzed on a BD FACS Canto II (BD Biosciences, San Jose, CA, USA).

*Immunofluorescence staining and confocal imaging.* Kidneys were fixed in 4% PFA overnight at 4 °C, washed with PBS, embedded in paraffin blocks, 5 µm sections

were generated and blocked in 10% normal donkey serum in PBS containing 0.3% Triton-X (blocking buffer) at room temperature for 1 h. Primary antibodies used included chick anti-GFP (1:250; Aves Labs), and rabbit anti-phospho-S6 ribosomal protein Ser240/244 (1:500; Cell Signaling Technology, Danvers, MA, USA, Catalog #5364). Slides were incubated with primary antibody in blocking buffer overnight at 4 °C, and washed 3 times in PBS with 0.1% Tween-20 for 20 min at room temperature. After washing, slides were incubated with secondary antibody (Alexa Fluor 488 donkey anti-chicken; 1:250; Jackson ImmunoResearch, Catalog #703-545-155 and Cy3 donkey anti-rabbit IgG; 1:250; Jackson ImmunoResearch) at room temperature for 1 h followed by washing and DAPI staining at 4 °C. Slides were mounted in Prolong Gold Antifade Reagent (Cell Signaling Technology) and imaged with a confocal Nikon A1R inverted microscope with a 20X objective lens using a pinhole of 1.2 µm on all channels at a resolution of $1024 \times 1024$.

*In vitro glycolysis and mitochondrial respiration assays.* FACS-isolated P0 NPCs (isolated as per the protocol above) were cultured for expansion for 1 week in NPEM[85], seeded onto a 24-well XF assay plate in XF assay media and submitted for Seahorse analysis of glycolytic activity and mitochondrial respiration per manufacturer's protocols for the XF Glycolysis Stress Test and XF Cell Mito Stress Test (Agilent Technologies, Santa Clara, CA, USA). ECAR and OCR measurements were collected for 2–3 replicate wells per sample and 3–4 samples per genotype.

*Pimonidazole injection and staining.* Postnatal day 0 (P0) pups were injected with 60 mg/kg pimonidazole (Hypoxyprobe^TM, Burlington, MA, USA) in sterile PBS; mice were sacrificed after 90 min and kidney tissue was immediately collected and fixed in 4% PFA overnight at 4 °C. Staining was performed with 1:50 dilution used for the pimonidazole-specific primary and FITC secondary antibodies; chick anti-GFP (1:250; Aves Labs) and guinea pig anti-cytokeratin 8 (1:250; Abcam, Cambridge, MA, USA, Catalog #ab194130) were used as described above with secondary antibodies as indicated (Alexa Fluor 647 goat anti-guinea pig; 1:250; ThermoFisher Scientific, Catalog #A-21450; and Cy3 donkey anti-chicken IgY; 1:250; Jackson ImmunoResearch, Catalog # 703-165-155). Quantification in GFP+ regions normalized to signal in adjacent tubules was performed in ImageJ.

*RNA in situ hybridization (RNAscope).* RNAscope for *Axin2* transcripts was performed per manufacturer's protocols for the RNAScope Multiplex Fluorescent Reagent Kit v2 Assay (Advanced Cell Diagnostics, ACD, Newark, CA, USA) for P0 and P2 paraffin sections generated as for immunofluorescence. Transcripts of *Axin2* were detected using a manufacturer-supplied probe (ACD 400331). Slides were stained with anti-Six2 (1:300; Proteintech, Rosemont, IL, USA; Catalog #11562-1-AP), anti-cytokeratin 8/18 (1:250; Abcam) and DAPI (1:500), mounted in Prolong Gold Antifade Reagent (Cell Signaling Technology) and imaged with a confocal Nikon A1R inverted microscope with a 40X objective lens using a pinhole of 1.2 µm on all channels at a resolution of $1024 \times 1024$. Images were processed in Imaris 9.6.0 (Bitplane, Zurich, Switzerland) using the spot function to count transcripts (puncta) in Six2+ surface objects (volumes). Thresholds were set such that there was ≤1 spot detected in negative control images. Statistical analysis was performed with a one-way ANOVA with Tukey's test for multiple comparisons; $p < 0.05$ denotes statistical significance.

*Surface imaging.* Postnatal day 4 (P4) kidneys were dissected in cold PBS and GFP was visualized using a Lecia stereo-fluorescent microscope with an exposure time of 6.9 s, gain of 1.3x, saturation of 1.15 and gamma of 0.51. The images were imported into Aperture 3.6 and batch-processed to remove green haze.

*RNA isolation and qPCR.* RNA from tissue and cells was extracted using the PureLink RNA Mini kit (Invitrogen) and cDNA was synthesized with SuperScript II reverse transcriptase (Invitrogen) following the manufacturer's instruction. Quantitative PCRs were performed using iTaq Universal SYBR Green Supermix (Bio-Rad, Hercules, CA, USA) on the StepOnePlus RT PCR system (Thermo-Fisher). Data were analyzed using the Delta-Delta-CT method. Oligos used are provided in Supplementary Table 1. Statistical analysis was performed with a two-tailed Student's t-test; $p < 0.05$ denotes statistical significance.

*Western blot.* Liver protein samples were lysed and extracted in Laemmli buffer, heated, and fractionated on an acrylamide gel. After transfer to a nitrocellulose membrane the proteins were detected with rabbit anti-Tmem59 antibodies at 1:1000 dilution (ThermoFisher Catalog #PA5-21575, Abcam Catalog #ab97597, or ProteinTech Catalog #24134-1-AP) using a BioRad ChemiDoc touch imaging system.

*Nephron counts.* HCl maceration of whole kidneys was performed according to[62,86]. Comparing the counts of CD1 kidneys at different ages (Supplementary Fig. S1D) established that stable nephron counts were obtained after P24, possibly reflecting the age when glomeruli become refractory to HCL lysis. Kidneys were isolated from ≥P28 animals, the capsule removed, minced using a razor blade and incubated in 6 N HCL for 90 min. The dissociated kidneys were vigorously pipetted every 20–30 min to further disrupt the kidneys. After incubation, 5 volumes of distilled water were added to the samples followed by incubation at 4 °C overnight.

**Table 1 Number of barcodes and percentage of reads mapped for each 10x scRNA-Seq sample.**

| Sample | Number of barcodes (cells) | Percentage of reads mapped (%) |
|---|---|---|
| E14KI | 7606 | 67.6 |
| E14TGC | 3643 | 66.5 |
| E14TSC | 3076 | 66.4 |
| P0KI* | 2032 | 39.0 |
| P0TGC | 3298 | 66.9 |
| P0TSC | 2473 | 67.1 |

*This sample was sequenced together with a rhesus sample, which does not map to the mouse genome.

Then, 100 μl of the macerate was pipetted into a cell-culture dish and glomeruli were counted in triplicate (three aliquots) for each sample. A single experimenter (A.E.J.), blinded to the genotypes of kidneys being scored, performed all counts. Genotyping was performed after the counts on DNA isolated from the liver, spleen, or tail using the primer sets reported in Supplementary Table 1. One-way ANOVA with Tukey's multiple comparisons tests were performed in GraphPad Prism version 8 to evaluate statistical significance. Both kidneys were counted for each animal with males and females represented.

**Quantification and statistical analysis**. All the parameters used in the analysis of the samples are reported in the scripts and are available on GitHub (see 'Code availability' section).

*Translatome data analysis*. RNA-Seq data were analyzed using AltAnalyze using Kallisto to map transcripts to Ensmart72/mm10. Gene expression was computed as log2-transformed transcripts per million (TPM) and the significance of differential expression was set to a fold change of >1.2 and a raw *p*-value of <0.05. Visualization of the data was performed with Morpheus https://software.broadinstitute.org/morpheus/.

*Processing raw scRNA-Seq data*. Raw scRNA-Seq data were sequenced using 10x Genomics v2 chemistry and was processed using Cell Ranger (v2.0.0) (https://github.com/10XGenomics/cellranger). Raw reads per cell were mapped to the mm10 mouse genome. As a quality-control metric, barcoded cells with less than ~5000 UMIs were dropped from the analysis, and for 5 out of 6 samples, ~66% of the reads were confidently mapped to the transcriptome with median of 2500 genes expressed per cell in each sample (the other sample was combined with an unrelated rhesus sample which does not map to the mouse genome, lowering the % of mapped reads). Finally, mapped reads were used to generate matrix of raw counts of genes across barcodes for each sample. Please refer to the Table 1 for number of barcodes and percentage reads mapped per sample.

*Quality filtering, dimension reduction, and clustering*. Seurat (v2.3.4) (https://doi.org/10.1016/j.cell.2019.05.031) package in R was deployed to perform quality control, selection of highly variable genes (HVGs), dimensionality reduction, and clustering. Briefly, cells with at least 100 detected genes and any gene expressed in ≥3 cells were used in subsequent analyses. For quality control, cells with ~3-fold higher than the median number of transcripts/cell (likely doublets) or cells with ~3.5-fold higher than the median mitochondrial gene fraction were excluded. Log2 transform followed by scaling was performed to normalize the raw counts of genes across all cells. Using normalized counts, cell cycle effect on clustering was minimized by regressing the difference between G2M and S phase. Next, HVGs were selected by marking the outliers from dispersion vs. avgEXP plot. These HVGs were then used to perform principal component analysis (PCA), a linear dimensionality reduction approach. Genes included in the first 20 principal components (PCs) were then used to cluster the cells, applying a graph-based clustering algorithm embedded in Seurat. Cell clusters were visualized by using both uniform manifold approximation (UMAP) and t-distributed stochastic neighbor embedding (tSNE) (a non-linear dimensionality reduction approach) and markers of each cluster were obtained using Wilcoxon rank-sum test in 'FindAllMarkers' function.

*Cell-type annotation*. Clusters were annotated to their respective cell-types using a priori knowledge of cell-type-specific markers and a computational framework. GSEA (https://doi.org/10.1073/pnas.0506580102, https://www.nature.com/articles/ng1180) of our cluster markers against gene-sets defined in Combes et al.[49] assisted in the computational and statistical annotation of cell clusters. A matrix of FDR of enrichment of gene sets in cell-clusters was extracted from the GSEA results using a custom python script (available on GitHub).

*Integrating samples to understand cell-type-specific responses*. To determine cell-type similarities and differences among the samples, we implemented the integration analysis provided in Seurat (v3.2.3). Quality filtering, normalization and cell-cycle effect regression was carried out as described above. Instead of selecting outliers from dispersion vs. avgExp plot, HVGs were determined using the variance stabilizing transformation 'vst' function which yielded approximately 2000 HVGs. Integration of the samples was performed by first identifying anchors using 'FindIntegrationAnchors', followed by integrating the datasets using anchors with 'IntegrateData' function. Dimensionality reduction was performed by PCA followed by both UMAP and tSNE. Graph-based clustering was used to generate clusters. Seurat built-in functions were used to generate dot plots, feature plots and heatmaps. Photoshop was used to merge single color feature plots into the multi feature image shown in Figs. 1Ci and 2B, C.

*Mutual nearest neighbor correction*. To systematically compare our data to published single-cell RNA-Seq data we applied mutual nearest neighbor (MNN) correction using 'FASTMNN' function in Seurat (v3.2.3). From corrected dataset MNN pairs were obtained to evaluate cell-type similarity.

*Functional and pathway enrichment*. To determine the biomarkers, pathways, and transcription factors enriched in our cell-types, we utilized GO-Elite[53], which can identify a minimal non-redundant set of biological functions and pathways enriched for a particular set of genes or metabolites. Markers identified by our Seurat analysis for each cluster were also analyzed by GO-Elite using the following parameters: 1.96 as *z*-score cut off for pruning ontology terms, Fisher's exact test for over-representation analysis (ORA), 0.05 as permuted *p*-value, at least 3 genes changed and 1000 permutations for ORA. The full GO-Elite output is presented in Supplementary Data 2 and 8.

*CellChat*. Seurat objects for each sample were merged by age to generate an E14 and P0 object for analysis of signaling pathway interactions suggested by the transcriptional signatures via the R package CellChat (v1.0.0)[56], implemented with default parameters; the script is available on GitHub.

*cellHarmony*. To evaluate the predominant global pattern of differential gene expression changes across all cell populations between $Six2^{KI}$, $Six2^{TGC;Tsc1}$, and $Six2^{TGC}$, the scRNA-Seq data were additionally analyzed using the software cellHarmony in AltAnalyze version 2.14. To focus on maximally reproducible changes, each single-cell capture was subdivided into three pseudo-bulk replicates, consisting of: (1) male, (2) female, and (3) unassigned cells based on the expression of sex-specific transcripts (https://git.io/Jlevy). cellHarmony was run on these pseudo-bulk replicates (Seurat-based cluster assignments for all cells) for all cell-type-specific comparisons (fold > 1.2 and empirical Bayes moderated *t*-test *p* < 0.05) using the PathwayCommons database in AltAnalyze's Ensembl version 72 database for gene set enrichment. cellHarmony results for different comparisons were combined into a single aggregate, organized UMAP using the cellHarmony-Combine function of AltAnalyze (https://git.io/J4RR7).

**Reporting summary**. Further information on research design is available in the Nature Research Reporting Summary linked to this article.

## Data availability
The RNA-sequencing data generated in this study have been deposited in NCBI's Gene Expression Omnibus (GEO) (Edgar et al.[87]) database under accession codes GSE173264 (https://www.ncbi.nlm.nih.gov/geo/query/acc.cgi?acc=GSE173264), GSE173265 (https://www.ncbi.nlm.nih.gov/geo/query/acc.cgi?acc=GSE173265), and GSE173266 https://www.ncbi.nlm.nih.gov/geo/query/acc.cgi?acc=GSE173266), and contain all processed bam files, raw counts of genes across barcodes/cells and cell annotations in the form of a metafile. Source data are provided with this paper.

## Code availability
All computer codes and scripts [R scripts, custom Python/Perl scripts and software packages] have been uploaded to GitHub [https://github.com/praneet1988/SingleCellAnalysisScripts][88] and Zenodo (https://doi.org/10.5281/zenodo.5525529)[89]. All codes have been licensed with a general public license (GPL v3.0).

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

## Acknowledgements

We would like to thank the Gene Expression Core, Dr. Yueh-Chiang Hu and the Transgenic Animal and Genome Editing Core facility, the Research Flow Cytometry Core (supported by NIH S10OD023410), and the Confocal Imaging Core at Cincinnati Children's Hospital Medical Center. We thank Dr. John Cobb for the generation of the *Rspo3*^f/f mouse line. We also acknowledge funding support from the National Institute of Diabetes and Digestive and Kidney Diseases (NIH DK106225 to R.K.; F30 DK123841 to A.E.J.), as well as generous support from an internal General Funds award at Cincinnati Children's Hospital Medical Center.

## Author contributions

R.K. and E.W.B. conceived the project. E.W.B. and A.E.J. designed and conducted experiments. R.K., E.W.B., A.E.J. and P.C. interpreted the data; N.S., P.C., A.E.J., E.W.B., and R.K. performed bioinformatic analysis of RNA sequencing data. R.K., A.E.J. and E.W.B. wrote the paper with input from all authors.

## Competing interests

The authors declare no competing interests.
