## [Peer Review File. · Nature Communications]

REVIEWER COMMENTS

Reviewer #1 (Remarks to the Author):

The cessation of nephron development in the metanephric kidney is an important process to understand from basic biology as well as regenerative medicine angles. In this manuscript Brunskill et al provide important new insights in this process using two different mouse models with increased nephron endowment, one of them the Six2GCE knock-in, which is itself an inactivating allele thus effectively halving the dose of Six2, the other a heterozygous conditional Tsc1 knockout driven by a Six2GC BAC transgene giving Cre activity in the same cells but with the normal Six2 dose. The BAC transgene alone provided the control genotype. The authors analyze E14 NPCs (young) and P0 NPCs (old) via scRNA-seq and Translating Ribosome Affinity Purification (TRAP), which gives a better representation of the protein expression in cells. The main conclusion of the work is a model where old NPCs shift towards final differentiation through an increased sensitivity to Wnt9b-mediated canonical Wnt signaling, thus increasing the level of beta-catenin signaling resulting in an shift from NPC self-renewal to differentiation.

Overall I love this manuscript and it certainly is for me the level of scientific interest and elegance I would expect a Nature Comm paper to have. Especially the use of the TRAP technique is enlightening. I do however have some issues the authors should address before I can support publication of the work in this journal.

1. A big first part of the manuscript describes the generation and analysis of the scRNA-seq data, and its comparison to comparable data from Combes et al. First, although I can (just about) follow this description of the analysis and all the different algorithms used, I do not consider myself expert enough on this aspect of the manuscript to have a justifiable opinion about the correctness of the analysis. I hope another reviewer will be more expert on this.

But apart from this, or maybe because of this, I find these almost 6 pages very detailed description too much for a general audience journal like Nature Comm, and as I result I think that (at least for me) the conclusions are getting a bit lost. I can imagine a more focused description of the conclusions of this part and details in supplementary information (because I agree it should be available in this level of detail) might be more appropriate for big parts of the audience. I would however be the first to admit this might be because of me being less expert on this part, and this clearly is a call for the editor to make.

2. For the single cell analysis there is an extensive explanation of the importance of the followed procedure to be able to exclude FACS sorting partially because of the cell stress this causes. Yet the TRAP analysis is dependent on pretty complex FACS isolation scheme. I understand why this is essential for this technique, but I think some discussion about this, and whether or not this affects the results would be useful.

3. I hate to be reviewer three who asks for ridiculous long extra experiments but the (very elegant) model coming from the data on increased sensitivity for Wnt9b in the shift from self-renewal to differentiation leaves some important gaps that could provide much more supportive evidence for this model. At present the confirmation is in my opinion pretty indirect, based on expression changes that would predict increased response to Wnt9b. But the key experiments, i.e. what is beta-catenin actually doing, are missing. This includes:

a. Is it possible to actually detect increased beta-catenin signaling? This could be done with beta-catenin activity reporter mice, or maybe phosphor/non-phospho specific beta-catenin antibodies. Could something be done with isolation of the cells with short-term culture and infection with beta-catenin activity reporter lentiviral constructs (i.e. PMID 20186325)? I realize the culture conditions

might require beta-catenin activating compounds making this problematic. How do known or predicted beta-catenin target genes behave in the expression data?

b. A way to functionally test the model could be to genetically increase or decrease beta-catenin signaling in the NPCs at the correct stage. The former could be done to use the Six2GCE driver the authors use anyhow to activate the conditional exon 3 activated beta-catenin mouse (PMID 10545105) and time the tamoxifen induction to specifically hit old NPCs. The latter could be done by looking at heterozygous beta-catenin knockout mice, which has been shown to reduce beta-catenin activity in Apc-mutant mice (PMID 23045279).

I realize these are big experiments and I happily leave the decision if things like this are required or not to the editor.

4. My biggest issue by far is with the generation and analysis of the Tmem59 knockout for the following reasons:

a. As I understand the description of what has been done, then the analysis was completely based on the first founders born from the zygote electroporation. For me, and many with me, analyzing CRISPR-edited founders is an absolute no-no. Even with zygote electroporation as used here the risk of mosaic animals is simply too big. Multiple alleles might be present in each animal (and Sanger sequencing will not always pick them up), there could be complete unedited cells in the animal, etc.

b. Using CD1 outbred mice will lead to even more variability from one animal to the next.

c. PCR and Sanger sequencing is never sufficient to determine the exact zygosity of the animal. CRISPR/Cas9 can make surprisingly large deletions which are simply not picked up by PCR, so a heterozygous large deletion mouse will appear wild type. Different ways are available to give a much more complete analysis of the possible allele(s) in a mouse are available, for instance using ddPCR, and in the end nothing can beat a good old(-fashioned) Southern blot. It's not still the golden standard for nothing.

d. The description of the mutants is weird and incomplete. Two homozygous mice (line357)? The exact mutation should be given. Pseudo wild-type (line357)? If there is a 2 amino acid deletion the mice are not wild type and one simply cannot state that these would behave as wild type without actually showing that (independent from the observed phenotype, that's circular reasoning), especially if the actual 2 aa mutation isn't even given? Heterozygote wild type? I don't have a faintest idea what this would mean. The same holds true for the further details in table S6, hardly any information on the alleles analyzed is given there.

e. The sgRNA targets exon 4. Since this exon contains 153 nt (i.e. exactly 51 amino acids) any splicing around the mutant exon would lead to a leaky mutant. However, no data at all is shown confirming the mutation state (RNA or protein).

There is simply no way the data from these animals can be interpreted in any way, the fact that it fits with the model is not good enough. I have no problem with using animals for research, but if we do, it comes with scientific and moral obligations to do these experiments properly, and this is here not the case. I might be old-fashioned but a new animal model cannot be interpreted without being properly described and analyzed first. If the authors would want to keep the supportive data from this model in the manuscript they should breed at least one of the mutants to real heterozygous F1, and subsequently to homozygous F2, determine and describe the actual mutations properly, confirm the effect of the mutation at protein or RNA level and only then determine the phenotype.

Some minor issues:

5. The authors consistently refer to the models as hemizygous. I may be wrong but I think the term hemizygous is reserved for situations like genes on the X chromosome in males or randomly integrated transgenes where there is a different copy number from the start. I think in this case the alleles should be called haploinsufficient.

6. I was very much interested in the analysis the authors did with GO-Elite GenMAPP, however the reference given (line 156) only mentions that GenMAPP is no longer supported, with a link to GenMAPP-CS which appears to be broken and a link to GO-Elite where I next got lost in a lot of other links that were broken and other things I personally did not understand. Of course it is not the authors' responsibility that tools that are not under their control remain available online, but for the sake of reproducibility and FAIR data use this is far from ideal. Are there still other places where GO-Elite can be found that could be mentioned?

7. Although the references are there it would be good for many readers I think to explain in one or two sentences what the importance is of Rpl10a for the TRAP analysis. A one sentence explanation of the technique might be sufficient.

8. The enrichment procedure for the TRAP (line 299-301) looks correct but could be helped with a little schematic (sup figure would be fine) where the different markers are found in the developing kidney to support this enrichment scheme.

--

Reviewer #3 (Remarks to the Author):

In the current manuscript the authors have aimed to examine molecular pathways that leads to changes in nephron endowment in the Tsc+/- mice. The Kopan group has previously published the role of Tsc in nephron endowment. The team have used a variety of methods including single cell sequencing, translational profiling and variety of informatics methods. Finally, the team zooms in on Rspo3 and Tmem59 as likely mediators and use mouse genetic models. The key novelty of the work in the Rspo3 and Tmem59 data. Overall the work is comprehensive and contains important and novel findings that should be interesting for the readership

The authors have performed single cell RNAseq in the Six2Ki, Six2TG and TSC mice on E14 and P0, one mouse per condition.

1, A key limitation of single cell RNAseq is that it not able to resolve differences in closely related cell types, such as the developing mouse kidney. This is expected as cell differentiation is a continuous process. The data seems very "overclustered" it seems that the authors have tried to generate a large number of subclusters to match prior stages on nephron differentiation. I think the authors should show the number of PC's and the rationale for choosing such high resolution.

The authors have spent a huge amount and honorable efforts to match the data with prior publication by Combes. Unfortunately, both datasets seem overclustered therefore while general patterns of gene expression are stable the exact same data cannot be recaptured.

2, The actual cell cluster markers are not shown in the paper. The authors have used the Combes data as a reference set and compare the different identified clusters.

1, I am very sympathetic with the kidney developmental biology group, such as their heroic effort to match the different kidney development stage markers with the single cell data such as NPC, SSB and PTA, however, I am not convinced that the single cell data completely support the prior single marker-based results. For example NPC1, NPC2, SSB, and PTA does not separate on the TSNE (a,b,d,i). I suggest that the authors either show their anchor gene expression data as a violin plot for naïve NPCs by expression (Six2+ ; Cited1+), committed, primed (Six2+ ; Cited1- ; Wnt4+) and pre-tubular aggregates (PTA, Six2+ ; Cited1- ; Wnt4+ , Pax8+ ; Fgf8+). as previously described or the community should just admit that the kidney development is a continuous process and these prior stages cannot fully separate by the single cell data.

I think to support their claim that the single cell data segregates NPC, and SSB and PT, the they

should show the

3, The overall clustering, which the author claim "superclustering" seems stable between datasets.

4, The supplemental table is interesting, it shows the highly overlapping gene expression. It is not clear what NA means in supplemental table1, why those gene expression is not shown.

5, Figure1 labeling does not seem to be complete the difference between Aa and Ab subpanels are unclear. It is hard to understand the differences between Aa and Ab

6, Figure 2. There is no scale shows for the bubble plots and it is not clear whether or not the same scale was used for multiple analysis. The upper and lower panel differences are not labelled.

7, On figure 2 the authors make conclusion on the number of cells in each cluster, however the single cell dataset is relatively small and the single cell analysis suffers from uneven drop-out. I think claims made on cell fractions would need to be supported by an independent method. For example stain and quantification.

8, Another critical limitation of the single cell analysis is that batch and biological variations cannot be dissected as the batch represents both a technical and biological differences. This effect is particularly problematic for datasets such as the current work, where there no biological replicates were analyzed. I think statements made around fig2N etc must be confirmed by independent methods.

9, If the primary goal was to understand differences between 3 genotypes why did not the author simply perform a cluster specific differential expression?

10, The separation of the samples in the Velocito data is concerning and the authors should show convincing data that this is not a batch effect issue in here. Can this be recapitulated on a single sample? The authors should show separately Six2, Cited and Wnt4 feature plots for the velocity data. From this data it seems Hnf4a is not really expressed. Again, one need to keep in mind that this is also batch effect sensitive. The arrows cannot be seen on the figure due to its size and resolution.

11, Where is Tsc expressed in the single cell data? Where is its highest activity? Where is mTOR?

12, I think the TRAP data is very nice, albeit I do not agree with the justification. Both scRNA and TRAP analyzes gene expression so TRAP will not be able provide insight on protein expression.

13, I think the Wnt data is interesting. The Rspo3 data is interesting and it is nice to have the foxd1 as a control dataset, however the wnt level is likely to be very tightly balanced so maybe low and high wnt levels could be problematic.

14, We would need to know results of the Tsc Rspo3 het mice? Are these changes specific for Tsc?

15, The Tmem59 data is clearly the highlight of the manuscript. How does Tsc regulate Tmem59? Where is tmem59 expressed? How does Tmem59 lead to an increase in nephron number?

16, The resolution of the figures are too low and the labels cannot be read on figure1

--

Reviewer #4 (Remarks to the Author):

The authors in this study take advantage of genic animal models, mainly RNA-seq and bioinformatics tools to understand the mechanisms of nephrogenesis. The tipping-point hypothesis is intriguing but more evidence is needed to prove it, and more evidence is also needed to support their conclusions in the discussion.

Comments:

- The translome, which is different from transcriptome, is a more accurate approximation for estimating the expression level of some genes. The advantage of TRAP technique designed for animal studies is that TRAP allows affinity purification of ribosome-bound mRNA directly from the- targeted tissues for further analysis of gene expression and identification of new transcriptomes and translomes. This may avoid potential artifacts resulting from tissue fixation, dissociation of tissues, or isolation of single cells from tissues when the entire translated mRNA complement of specific tissues is examined from intact tissues in vivo. Thereby, if the isolated cells from transgenic animals as described in this manuscript are used for TRAP and RNA-seq, the value will be decreased or limited due to the changes in the ribosome-bound RNA resulting from the isolation process.
 - It is not clear how the studies in result section 1 support the statement "The cellular composition of the nephron progenitor niche is unaffected by mutations ...",
 - More biological evidence is needed to support the "tipping point" hypothesis. General description that multiple pathways and mechanisms may responsible for the gene expression differences that separate the indicated genotypes appears not sufficient as a suggestion. What possible pathway and mechanisms are involved? Could the authors show the phenotype evidence of different stages for the mice with different genotypes?
-
- It could be helpful to provide cellular and molecular evidence that cell population switches from self-renewal to differentiation.

Minor concerns:

1. In lines 189-198, authors should discuss these changes in more detail. Furthermore, further discussion on the differences between the rest of the genes analyzed should be added.
2. What is reason why the authors decided not to evaluate the Six2KI genotype for translome analysis (TRAP). In line 245 authors mention that "the increased nephron endowment in Six2KI might be driven by a mechanism acting earlier in nephrogenesis, before P0". It would be interesting to verify with this genotype if there is any change in the translome with respect to the other two genotypes.
3. In the introduction section, authors should mention a little more about Wnt pathway and its relation to the nephron endowment/nephrogenesis. Since the authors focus most of the results in this pathway.
4. In line 50 "nephrogenesis ends by P3", the meaning of P3 should be added.
5. In line 64 "and can and contribute nephrons", please remove the second word "and" from the sentence.
6. In line 78 "transcriptome at two developmental timepoints (E14 and P0) of three genotypes", the meaning of E14 and P0 should be added.
7. In lines 139 and 148 specific reference should be added.
8. In Fig 1B and Fig 2M, the image resolution should be improved.
9. Reference to Fig 3C is missing in the text, please include it where it belongs.
10. In line 495, the concentration of RNase inhibitors and protease inhibitors should be included.
11. In line 531, please complete the reference with the name of the authors.
12. The authors studied E14 and P0 but in the method section for protein quantification they stated that kidneys from E14 or P0, P1, or P2 mice were isolated in ice-cold PBS and the capsule removed.

We are extremely grateful for the thoughtful review provided by the referees. All made incredibly valuable observations and comments exposing interpretations for which we could not generate orthogonal support. This prompted us to seek independent experimental data to either impeach or support our central conclusion. We are happy to report that our central conclusions are still supported. Our detailed response is below each comment.

Referee #1 (Remarks to the Author):

The cessation of nephron development in the metanephric kidney is an important process to understand from basic biology as well as regenerative medicine angles. In this manuscript Brunskill et al provide important new insights in this process using two different mouse models with increased nephron endowment, one of them the Six2GCE knock-in, which is itself an inactivating allele thus effectively halving the dose of Six2, the other a heterozygous conditional Tsc1 knockout driven by a Six2GC BAC transgene giving Cre activity in the same cells but with the normal Six2 dose. The BAC transgene alone provided the control genotype. The authors analyze E14 NPCs (young) and P0 NPCs (old) via scRNA-seq and Translating Ribosome Affinity Purification (TRAP), which gives a better representation of the protein expression in cells. The main conclusion of the work is a model where old NPCs shift towards final differentiation through an increased sensitivity to Wnt9b-mediated canonical Wnt signaling, thus increasing the level of beta-catenin signaling resulting in an shift from NPC self-renewal to differentiation.

Overall I love this manuscript and it certainly is for me the level of scientific interest and elegance I would expect a Nature Comm paper to have. Especially the use of the TRAP technique is enlightening. I do however have some issues the authors should address before I can support publication of the work in this journal.

1. A big first part of the manuscript describes the generation and analysis of the scRNA-seq data, and its comparison to comparable data from Combes et al. First, although I can (just about) follow this description of the analysis and all the different algorithms used, I do not consider myself expert enough on this aspect of the manuscript to have a justifiable opinion about the correctness of the analysis. I hope another reviewer will be more expert on this. But apart from this, or maybe because of this, I find these almost 6 pages very detailed description too much for a general audience journal like Nature Comm, and as I result I think that (at least for me) the conclusions are getting a bit lost. I can imagine a more focused description of the conclusions of this part and details in supplementary information (because I agree it should be available in this level of detail) might be more appropriate for big parts of the audience. I would however be the first to admit this might be because of me being less expert on this part, and this clearly is a call for the editor to make.

Point well taken. We have revised the text to give a broader perspective of the goals of each analysis, and have moved technical details to the methods. We previously provided a very technical description to address important technical/analytical challenges that are commonly faced in single-cell informatics. Specifically, we completed this comparative analysis to address three issues: 1) evaluate for batch effect to overcome the need to pool independent biological replicates (each of our samples does represent a pool of multiple animals), 2) test the hypothesis that NPC from strains with more nephrons are discernable from controls based on their transcriptome, and 3) identify whether accumulation of uncommitted progenitors in strains with more nephrons can explain this increase. We restructured the text to get more quickly into the meat of the paper and moved some of these analyses to supplementary information. We have accomplished the first point in the revised manuscript by performing Mutual Nearest Neighbor based correction and found satisfactory agreement of our clustering results for Six2CE (Six2KI in the manuscript) with age matched sorted NPC of the same strain (Combes et al). See new Supplementary Figure 1C (below) which, along with the previously performed orthogonal GSEA, confirm that our dataset captured all cell types in the developing kidney cortex revealed by published datasets.

Supplementary Figure 1. (C): UMAP of E14 Six2KI cortically enriched kidney cells from 3 pooled embryos established 14 major clusters and 4 minor ones. 6 clusters formed the NPC "continent", and 7 formed the stromal continent. Nearest Neighbor based correction assigned cells to the NPC clusters seen in Combes et al. All cells in NPC cluster 1 were correlated with Combes,

~47% with C1-CC. 31% of NPC cluster 0 correlated with Combes C0, the rest with C2-CC. Similar results were obtained with GSEA, establishing correlation with FDR score <0.05 (B). “% Highest assigned cells in cluster” shows how many cells in our clusters were assigned to an individual Combes cluster. “%cells assigned to all clusters” shows how many cells in our clusters were assigned to any of the Combes cluster (Note that cluster 14 cells were assigned to multiple Combes clusters, resulting in 109% assigned cells).

2. For the single cell analysis there is an extensive explanation of the importance of the followed procedure to be able to exclude FACS sorting partially because of the cell stress this causes. Yet the TRAP analysis is dependent on pretty complex FACS isolation scheme. I understand why this is essential for this technique, but I think some discussion about this, and whether or not this affects the results would be useful.

We will discuss the caveat of TRAP on FACS-isolated cells (see also our response to Reviewer 4). There are important differences in the goals and methodology of the two technologies, which explains the differences you note. Although we removed *Tsc1* only from NPC, we anticipated gene expression changes in stroma and UB cells in response to the changes in NPC. To survey possible cell type responses to *Tsc1* loss, we collected all cortical cell types, using scRNA-seq. We used best available methodologies and practices to minimize stress (cold protease, and minimization of time from isolation to 10X Genomics barcoding). For the translome, we focused on the NPC and tested an available Notch2-TRAP as well as creating a Cited1-TRAP to look at naïve NPC directly. Unfortunately, the NPC expression of GFP in both was too low to detect (as in Figure R1). Thus, we opted for the strategy described in the paper and applied FACS to purify the NPC (cell of interest) from all other nephron cell types which, being descendants of the NPC, also contain the Rpl10a-EGFP fusion protein. To minimize cell stress during sorting, selection of an appropriately wide flow cytometer nozzle (100µm), and precoating of collection tubes with buffer solution were applied. We did not see a stress signature in the translome, suggesting that this compartment is less sensitive to the digestion conditions and sorting than the transcriptome. It is important to note that cycloheximide was added to faithfully freeze elongating ribosomes in their in vivo translational position during cell isolation and FACS. We agree that a discussion of this step is important and added it to the revised manuscript.

Expression ofEGFP-L10a in N2 BacTrapEGFP-L10 transgenic mice

Figure R1: Expression of EGP-RPL10a in N2 Bac Trap. Note absence from NPC, expression in epithelial descendants.

3. I hate to be reviewer three who asks for ridiculous long extra experiments but the (very elegant) model coming from the data on increased sensitivity for Wnt9b in the shift from self-renewal to differentiation leaves some important gaps that could provide much more supportive evidence for this model. At present the confirmation is in my opinion pretty indirect, based on expression changes that would predict increased response to Wnt9b. But the key experiments, i.e. what is beta-catenin actually doing, are missing. This includes:

a. Is it possible to actually detect increased beta-catenin signaling? This could be done with beta-catenin activity reporter mice,

We share your desire for better tools to access β -catenin activity. Unfortunately, the Axin2-reporter mice, for some unknown reason, do not label NPC, necessitating several studies that were dedicated to asking if canonical Wnt signaling was involved (for example, ¹⁻⁴). The same applies to phospho- β Cat or nuclear β Cat: I cannot recall any successful staining for these antibodies in the NPC (though they do work in the epithelial nephron, see ⁵). Of note, the transcriptome data indicate that Axin2 is unchanged between E14, P0, and P0TSC (Supplementary Table S6). As an orthogonal validation approach, we used RNAScope to detect Axin2 transcripts levels at P0 and P2 in *Six2*^{TGC} controls and *Six2*^{TGC;Tsc1} NPC as done in Fig 4 ⁶ by Schedl et al. Quantification of Axin2 transcripts (Figure 4E in revised manuscript, shown below) normalized to the volume of the Six2+ NPC population showed similar counts at P0 in TGC and TSC, in agreement with the transcriptome. At P2, however, Axin2 levels in *Six2*^{TGC;Tsc1} remained the same of P0 controls, while declining in the control *Six2*^{TGC}. It might suggest that at P2, *Six2*^{TGC;Tsc1} NPC more closely resemble the E14 and P0

population than P2 *Six2*^{TGC}. We noted that only *Rspo3*, *Tmem59*, *Lrp5*, *Wnt7b*, *Plaur* and *Sf3b5* Wnt pathway genes were (a) reduced in P0Tsc1 relative to P0, and (b) had an adjusted pValue <0.05 (as shown in Supplementary Table S6). From this list, based on expression level and pattern, and expected function we selected *Tmem59* and *Rspo3* for further validation. We report that *Rspo3* is a likely effector of *Tsc1* hemizyosity in NPC. Combined, we suggest that overall Wnt activity was not impaired, but we propose that causative Wnt targets may be present among the P0Tsc1 translated mRNA that are similar to E14 and different from P0. We are addressing their identity in a subsequent effort.

Figure 4E: Total counts of Axin2 probes normalized to volume of nephrogenic niches

or maybe phosphor/non-phospho specific beta-catenin antibodies.

This was not very successful in NPC, but see ⁵ for staining in nascent nephron epithelia. We too failed to detect a nuclear signal using the latest imaging technologies available to our

imaging core.

Could something be done with isolation of the cells with short-term culture and infection with beta-catenin activity reporter lentiviral constructs (i.e. PMID 20186325)? I realize the culture conditions might require beta-catenin activating compounds making this problematic. How do known or predicted beta-catenin target genes behave in the expression data?

Based on our PCA analysis of published data⁷, the profile of sorted cells differentiates young NPC from older ones. By contrast, the cultured cells are neither “old” nor “young”, they have a unique “culture” signature regardless of age at isolation or time in culture (See Figure R2). Thus, it will be difficult to interpret the data generated *in vitro*. Lentivirus infectivity in NPC (*in vivo* or *in vitro*) is very poor in our hands.

Figure R2. PCA analysis of published bulk RNA-seq data⁷ obtained from young (iE11.5, iE13.5) and old (iE16.5, iP1) NPC cultured for 20 or 80 days in “niche” medium (20c or 80c in the figure), compared to freshly sorted young (E11.5, E12.5, E13.5) and old (E16.5, P0) NPC. Note that this data confirms an observation we published in 2015⁸ that age has a differentiating molecular signature factored in PC2, whereas *in-vivo* vs. *in vitro* is factored in PC1. The accession number for the RNA-seq data reported in [7] is GEO: GSE78772.

b. A way to functionally test the model could to by genetically increase or decrease beta-catenin signaling in the NPCs at the correct stage The former could be done to use the Six2GCE driver the authors use

anyhow to activate the conditional exon 3 activated beta-catenin mouse (PMID 10545105) and time the tamoxifen induction to specifically hit old NPCs.

Some of what is suggested was published by others¹⁻⁴). In brief, activating bCat leads to early and/or abnormal differentiation, and may be a supraphysiological effect. Note also that titrating Wnt signals (via ligand, agonist, or antagonist addition) showed that low levels of bCat support self-renewal and high levels promote differentiation². Here, we show that *in vivo*, the differences in the translome, evolving with time and dependent on *Tsc1*, are able to create an activity gradient for specific transcripts but overall, we cannot claim that Wnt9b signals were down across the board. We now state on Page 12:

“Importantly, key Wnt agonists were among the transcripts reduced in the translome in P0 *Six2*^{TGC;Tsc1} relative to P0 *Six2*^{TGC} NPCs (blue box, Figure 4B): *Rspo1*, and *Rspo3*; required to engage *Znrf3/Rnf43* and prolong the half-life of the Wnt/Fzd complex, and *Tmem59*, coding for a molecule thought to cluster Fzd proteins thus increasing avidity/signal strength of WNT signaling⁹. The combination of increased antagonists and decreased agonists was unique to the Wnt signal transduction pathway (Supplementary Figure S4C-D). Although many known Wnt targets were generally upregulated in P0 *Six2*^{TGC;Tsc1} NPC, quantification of Axin2 transcripts by RNAScope in NPCs at P0 and P2 revealed that while Axin2 levels decreased with age in control *Six2*^{TGC}, P2 *Six2*^{TGC;Tsc1} resembled the younger cells (Figure 4E), suggesting that classical readouts of Wnt activation may not reflect accurately NPC self-renewal and differentiation activities. This notion is further supported by translome analysis which shows that Axin2 was not differentially bound to RPL10 in E14 vs. P0 progenitor cells, despite the fact that young and old NPCs display different self-renewal/replication rates¹⁰. Combined, these observations suggest the hypothesis that hamartin/Tsc1 regulates some aspects of the Wnt9b signal by affecting the translation of key transcripts, and the impact on nephrogenesis in *Tsc1* hemizygotes might be due to altered perception of Wnt signal input to below the threshold required for completion of MET, delaying NPC exit from the niche as well as cessation of nephrogenesis”.

The latter could be done by looking at heterozygous beta-catenin knockout mice, which has been shown to reduce beta-catenin activity in Apc-mutant mice (PMID 23045279). I realize these are big experiments and I happily leave the decision if things like this are required or not to the editor.

As we state above, and cited in the text, others have shown that titrating canonical Wnt signals will impact the choice between self-renewal and differentiation. Confounding the suggested experiment is the finding that incoherent feed forward logic makes Wnt signaling insensitive to b-Catenin dosage¹¹. Nonetheless, we performed the experiment by deleting a floxed allele of *Ctnnb1* with *Six2*Cre. We see no difference in nephron number in NPC-specific bCat hemizygosity (Figure R3).

Figure R3: Nephron number, measured via the acid maceration method, from kidneys isolated from littermates at P28. Note the ~30% reduction in nephron number due to the presence of the *Six2* BAC transgene (as we previously reported¹²). NPC-specific Beta-catenin hemizygotes are indistinguishable from *Six2* TGC controls littermates.

4. My biggest issue by far is with the generation and analysis of the *Tmem59* knockout for the following reasons:

a. As I understand the description of what has been done, then the analysis was completely based on the first founders born from the zygote electroporation. For me, and many with me, analyzing CRISPR-edited founders is an absolute no-no. Even with zygote electroporation as used here the risk of mosaic animals is simply too big. Multiple alleles might be present in each animal (and Sanger sequencing will not always pick them up), there could be complete unedited cells in the animal, etc.

We agree that the description of the *Tmem59* mutants left much to be desired, and have extensively revised it.

We note that our initial data presented nephron counts from both F0 CRISPR founders and F1 compound mutants generated from matings between two different founders. Nonetheless we completely agree that purification of the mutant *Tmem59* alleles in the F2 generation is needed. To that end, we selected three mutations (13bpDEL, 69bpDEL and 822bpDEL) to analyze (revised Supplementary Figure 5, replacing data based on F0/F1 generations). Wild-type outbred CD1 mice were used to generate these mice, and nephron phenotyping is performed while blinded to the genotype of the animal. Importantly, we neglected to pay attention to the fact that *Tmem59* was reduced, not lost, in the *Tsc1* hemizygote NPC. Therefore, we asked in the revised manuscript if heterozygote *Tmem59* synergize with *Rspo3* (they do not). We concluded that the gain of nephrons in *Tmem59* mutants is immaterial to the mechanism of action downstream of *Tsc1*, and delegated this story to supplemental information.

Supplementary Figure 4F-G: Nephron number, measured via the acid maceration method, from kidneys isolated at P28.

b. Using CD1 outbred mice will lead to even more variability from one animal to the next.

We were concerned about this in the original manuscript, which is why we have presented data on the variability in inbred (C57Bl6) and outbred mice (CD1) in the original Supplementary Figure 1A. We found no difference in the magnitude of inter-strain variability, as shown. We were curious about this in the context of another project in the lab and acquired the most variable mice in existence from the Collaborative Cross (https://comp.gen.unc.edu/wp/?page_id=99). These are stable hybrids of 8 strains, three of which are isolated from the wild. While they all differ from each other, the variability between individuals within each of the 12 strains are remarkably low, and similar in scale, arguing for a strong genetic determinant setting NN. This data set is presented here for the reviewer and will be amended as Figure S1A.

Supplementary Figure 1A: Violin plots of nephron number (NN) variation among individuals of the same strain. NN were counted by acid maceration in animals older than p28 (after p25 glomeruli are resistant to acid, Supplementary Figure 1D). the number of animals tested is listed above each strain. Note the variance within the inbred C57Bl6 is not smaller than that seen in eight-parent strains from the collaborative cross (CC strains) or CD1. Note also that NN cluster along a high value (C57, CC003, 006,007,008,013, 021, 026) or a low value (CD1, CC021,045,051,071) independent of body weight (CD1 mice are larger than C57 mice).

c. PCR and Sanger sequencing is never sufficient to determine the exact zygosity of the animal. CRISPR/Cas9 can make surprisingly large deletions which are simply not picked up by PCR, so a heterozygous large deletion mouse will appear wild type. Different ways are available to give a much more complete analysis of the possible allele(s) in a mouse are available, for instance using ddPCR, and in the end nothing can beat a good old (-fashioned) Southern blot. It's not still the golden standard for nothing.

We agree that a large deletion may be undetectable by PCR (and indeed, we identified the large deletion with an alternate primer set). We focused the revised figure on F2 offspring of the 13bpDel, 69bpDel (skips Exon4), and large deletion (822bp) alleles, as advised by the reviewer. Note, however, as explained above, we determined that these mutants do not address the *Tsc1* mechanism.

d. The description of the mutants is weird and incomplete. Two homozygous mice (line357)? The exact mutation should be given. Pseudo wild-type (line357)? If there is a 2 amino acid deletion the mice are not wild type and one simply cannot state that these would behave as wild type without actually showing that (independent from the observed phenotype, that's circular reasoning), especially if the actual 2 aa mutation isn't even given? Heterozygote wild type? I don't have a faintest idea what this would mean. The same holds true for the further details in table S6, hardly any information on the alleles analyzed is given there.

We agree, and we revised this extensively. These F0 and F1 animals are no longer included and the analysis now includes WT, heterozygous and homozygous *Tmem59* F2 offspring of F1 (+/13bpDEL, +/822bpDEL, and +/69bpDEL) cross. The new Supplementary Table S7 includes all the sequences, and Supplemental Figure S5 now is based on F2 crosses from three independent lines. The new data includes the genotype for the selected founder allele, and F2 analyses demonstrating that no hidden deletions are present in Supplementary Figure S5.

e. The sgRNA targets exon 4. Since this exon contains 153 nt (i.e. exactly 51 amino acids) any splicing around the mutant exon would lead to a leaky mutant. However, no data at all is shown confirming the mutation state (RNA or protein).

We agree, and we attempted to address this omission in the revised manuscript characterizing *Tmem59* in the lines we propagated (we included the data in the supplemental materials). Using three commercially-available *Tmem59* antibodies, several lysis and Western blot protocols, we could not detect protein loss or migration differences at the indicated protein size. This was unfortunate, so we hoped that nonsense mediated mRNA decay will enable us to demonstrate loss of *Tmem59* at the RNA level via qPCR for the C-terminal end of the transcript. We did identify a change in abundance of the c-terminal end, with cDNA sequencing across exon 4 confirming the animals are homozygous to the expected non-sense mutation. We are left with a confirmed change that should produce a truncated protein, but no validated antibody capable of detecting a change.

There is simply no way the data from these animals can be interpreted in any way, the fact that it fits with the model is not good enough. I have no problem with using animals for research, but if we do, it comes with scientific and moral obligations to do these experiments properly, and this is here not the case. I might be old-fashioned but a new animal model cannot be interpreted without being properly described and analyzed first. If the authors would want to keep the supportive data from this model in the manuscript they should breed at least one of the mutants to real heterozygous F1, and subsequently to homozygous F2, determine and describe the actual mutations properly, confirm the effect of the mutation at protein or RNA level and only then determine the phenotype.

We agree, and as detailed above we have done just that in the revised submission. We further note that given our findings that *Tmem59* mutations can alter NN but does not contribute to our *Tsc1* mechanism. We therefore added additional studies probing alternate mechanisms including metabolism/respiration status.

Some minor issues:

5. The authors consistently refer to the models as hemizygous. I may be wrong but I think the term hemizygous is reserved for situations like genes on the X chromosome in males or randomly integrated transgenes where there is a different copy number from the start. I think in this case the alleles should be called haploinsufficient.

We looked into this issue seeking some clarity for the definitions. A locus may be hemizygous (exist in one copy) and yet not be haploinsufficient. From the NCI dictionary of genetic terms (<https://www.cancer.gov/publications/dictionaries/genetics-dictionary/def/hemizygous>):

Hemizygous “Describes an individual who has only one member of a chromosome pair or chromosome segment rather than the usual two. Hemizyosity is often used to describe X-linked genes in males who have only one X chromosome. This term is sometimes used in somatic cell genetics where cancer cell lines are often hemizygous for certain alleles or chromosomal regions.”

Biology Online (<https://www.biologyonline.com/dictionary/hemizygous>) definition:

adjective

- (1) Characterized by having one or more genes without allelic counterparts.
- (2) Pertaining to a diploid cell with only one copy of a gene instead of the usual two copies.

For example, the male is hemizygous for most X chromosome genes.

Since we are removing one allele, we called these mice hemizygous as they have “one gene without allelic counterparts”.

From the NCI source, haploinsufficient stands for:

The situation that occurs when one copy of a gene is inactivated or deleted and the remaining functional copy of the gene is not adequate to produce the needed gene product to preserve normal function.

One can argue that the *Six2*^{TGC^{+/tg}}; *Tsc1*^{+/f} allele is both hemizygous and, if its role is to accelerate NPC differentiation, likely haploinsufficient. The *Six2* allele in *Six2*^{+/K1} mice is hemizygous, but I am not sure we can call it haploinsufficient as all characterized *Six2* functions are fulfilled.

We welcome editor input in the selection of the correct nomenclature.

6. I was very much interested in the analysis the authors did with GO-Elite GenMAPP, however the reference given (line 156) only mentions that GenMAPP is no longer supported, with a link to GenMAPP-CS which appears to be broken and a link to GO-Elite where I next got lost in a lot of other links that were broken and other things I personally did not understand. Of course it is not the authors' responsibility that tools that are not under their control remain available online, but for the sake of reproducibility and FAIR data use this is far from ideal. Are there still other places where GO-Elite can be found that could be mentioned?

Thank you for catching this. We replaced the link to GO-Elite with a working link: http://www.genmapp.org/go_elite/

7. Although the references are there it would be good for many readers, I think to explain in one or two sentences what the importance is of Rpl10a for the TRAP analysis. A one sentence explanation of the technique might be sufficient.

We added a clarifying paragraph.

“Given the limitations of scRNA-seq (including discordance with the protein state of the cell¹³, and dropout of undetected transcripts), coupled with the intriguing distribution of *Rpl29* (Figure 3A), we shifted to an

alternative strategy of Translating Ribosome Affinity Purification (TRAP;¹⁴) which has proven to provide a transcript profile with high concordance to the proteome. Having failed to detect NPC expression in Notch2-TRAP (obtained from <http://www.gensat.org/>) engineered to contain the ribosomal protein L10a EGFP fusion (EGFP/Rpl10a), we pursued a strategy utilizing mice containing EGFP/Rpl10a preceded by a floxed stop cassette in front of the powerful CAGGS promoter¹⁵.”

8. The enrichment procedure for the TRAP (line 299-301) looks correct but could be helped with a little schematic (sup figure would be fine) where the different markers are found in the developing kidney to support this enrichment scheme.

Good idea! We added a diagram and gating strategy, Supplementary Figure 4A .

Referee #3 (Remarks to the Author):

In the current manuscript the authors have aimed to examine molecular pathways that leads to changes in nephron endowment in the *Tsc*^{+/-} mice. The Kopan group has previously published the role of *Tsc* in nephron endowment. The team have used a variety of methods including single cell sequencing, translational profiling and variety of informatics methods. Finally, the team zooms in on *Rspo3* and *Tmem59* as likely mediators and use mouse genetic models. The key novelty of the work in the *Rspo3* and *Tmem59* data. Overall the work is comprehensive and contains important and novel findings that should be interesting for the readership. The authors have performed single cell RNAseq in the *Six2Ki*, *Six2TG* and *TSC* mice on E14 and P0, one mouse per condition.

To clarify, each 10x sample contained a pool of 3-5 embryos into one sample, not a single mouse.

1, A key limitation of single cell RNAseq is that it not able to resolve differences in closely related cell types, such as the developing mouse kidney. This is expected as cell differentiation is a continuous process. The data seems very “overclustered” it seems that the authors have tried to generate a large number of subclusters to match prior stages on nephron differentiation. I think the authors should show the number of PC’s and the rationale for choosing such high resolution.

We needed to be more succinct and clearer as to the intent of this section. We have not made any a priori assumption about the number of clusters. We performed un-supervised clustering at a resolution of 0.8 with the algorithms described (unless specifically stated otherwise). Following the reviewer comment, we were concerned that indeed we biased the outcome of the analysis. We thus repeated the analyses at the resolution settings of 1.4, 1, and 0.5. Only one stromal cluster was split at the lower resolution (0.5), and two stromal clusters fused at higher resolution (1, 1.4). Importantly, all NPC clusters were unaffected within the 0.5-1.4 range, suggesting the differences are robust and insensitive to arbitrary choices of resolution setting. We take this as an indication that we did not over-cluster.

For the reviewers, we are providing a link to all the PC’s and TSNE plots at the different resolutions (<https://github.com/praneet1988/Age-Dependent-Changes-in-the-Progenitor-Translatome-Coordinated-in-part-by-Tsc1-Increase-Perception->). We also added UMAPs, but since we wished to compare to data published using Seurat 2.4 we will not use the UMAP in this figure.

The authors have spent a huge amount and honorable efforts to match the data with prior publication by Combes. Unfortunately, both datasets seem overclustered therefore while general patterns of gene expression are stable the exact same data cannot be recaptured.

We aimed to address three issues by these analyses:

(1) Specifically, evaluate for batch effect to overcome the need to pool independent biological replicates, (2) test the hypothesis that NPC from strains with more nephrons are discernable from controls based on their transcriptome, and 3) identify whether accumulation of uncommitted progenitors in strains with more nephrons can explain this increase. We restructured the text to get more quickly into the meat of the paper and moved many of these analyses to supplemental information. We have accomplished the first point in the revised manuscript by performing Mutual Nearest Neighbor (MNN) based correction and found satisfactory agreement of our clustering results for Six2CE (Six2KI in the manuscript) with age matched sorted NPC of the same strain (Combes et al). See new Supp. Figure 1C (in response to reviewer #1) which, along with the previously performed orthogonal Gene Set Enrichment Analysis confirm that our dataset captured all cell types in the developing kidney cortex revealed by published datasets. Finally, UMAP integration analysis of all 6 datasets we generated produced a temporally organized NPC cluster with no outliers. These analyses verified the robustness among scRNA datasets generated by two different groups with different technical and bioinformatic methods. Notably, one of their clusters, C14, was not detected in ours and is likely composed of NPC/str doublets. By finding such strong agreement between our data and theirs we conclude the batch effect did not undermine our interpretation. As this section is largely a QC exercise, we will reduce the amount of space dedicated to this comparison as requested by Reviewer #1.

We rejected, the second hypothesis (that NPC from strains with more nephrons are discernable based on their transcriptome) as the only new (minor) cell clusters found represented likely contamination (neurons, C20) or doublets (C17). Likewise, no accumulation of uncommitted progenitors is evident in strains with more nephrons to explain this increase.

2, The actual cell cluster markers are not shown in the paper. The authors have used the Combes data as a reference set and compare the different identified clusters.

We apologize for the confusion. We improved the introduction of table S1 and created a better ReadMe document to explain what it contains. All the markers for E14 Six2KI were/are shown in Tables S1. None of our tables contains data from Combes et al., we do however use their formatting, and their nomenclature to label cell types in our dataset. The top 20 markers for each cluster are presented on the summary tab in Tables S1, S4. All the markers for each cluster are listed in 17/20 tabs numbered by the cluster number (tab zero is cluster zero, etc.). The "Lookup" tab contains the top three markers for each cluster, mitochondrial genes (when not filtered out), and enables the reader to search the entire dataset by cluster for expression of any GOI (gene of interest) using Excel "VLOOKUP" function, which we explain at the top of this tab. In addition, Table S4 contains all the markers from all genotypes/ages from the integration analysis, sorted by cluster as in Table S1. Supplementary Figure 1E shows the individual samples in the integration with clusters corresponding to Table S4, and pooled by age (1F). Combined, the integration analyses can also be viewed as a test for batch effect, attesting for the similarities between the six samples.

1, I am very sympathetic with the kidney developmental biology group, such as their heroic effort to match the different kidney development stage markers with the single cell data such as NPC, SSB and PTA, however, I am not convinced that the single cell data completely support the prior single marker-based results. For example NPC1, NPC2, SSB, and PTA does not separate on the TSNE (a,b,d,i). I suggest that the authors either show their anchor gene expression data as a violin plot for naïve NPCs by expression (Six2+ ; Cited1+), committed, primed (Six2+ ; Cited1- ; Wnt4+) and pre-tubular aggregates (PTA, Six2+ ; Cited1- ; Wnt4+ , Pax8+ ; Fgf8+). as previously described or the community should just admit that the kidney development is a continuous process and these prior stages cannot fully separate by the single cell data.

Thank you for the suggestion- we added violin plots for the suggested markers by cluster (Supplementary Figure 2A). We also created UMAP for each- the UMAP preserves global structure of clusters so the biological

Supplementary Figure S2: Violin plots of Six2TGC (Top row), Six2TSC (middle) and Six2KI (bottom row) at E14 (left) and P0 (right). UMAP diagram added to identify clusters (NPC encircled in a red, dashed line). Gene name shown above the plots, and the clusters are ordered from left to right in the same order as they appear in the UMAP (top to bottom, enlarge to view). NPC clusters are also marked with red box near the cluster numbers. Note that at E14, *Six2+*; *Cited1+* (*Wnt4-*) clusters are present in both genotypes, and the same is true for C0 in P0 Six2TGC. By contrast, all NPC clusters at Six2TSC at P0 are *Six2+*; *Cited1+*; *Wnt4+* though C2 has the least amount of Wnt4. This is consistent with the data in Figure 2

difference among cell types becomes more evident. Note that UMAP correctly identify the continuum between naïve NPC and epithelia.

I think to support their claim that the single cell data segregates NPC, and SSB and PT, they should show the:

3, The overall clustering, which the author claim “superclustering” seems stable between datasets.

The UMAP integration data generated by Seurat 3 (Figure 1C, Supplementary Figure 1E,F, Table S4) show this - all the data set was well resolved by UMAP into “continents” comprised by stromal, UB, and NPC “territories”, respectively (with a few small “islands” for immune cells, etc.). Moreover, the integration UMAP analysis correctly aligned cells along a developmental pseudotime- Cap Mesenchyme (*Cited1+*) in one pole,

epithelia (Epcam+) in the other, closer to the epithelial UB. If batch and technical issues would have been strong confounders, outliers would have formed (For UMAP of individual cluster, see above inserted in the violin plots).

4, The supplemental table is interesting, it shows the highly overlapping gene expression. It is not clear what NA means in supplemental table1, why those gene expression is not shown.

We apologize again for the poor introduction to the Lookup tables. #N/A (not available) is the output of a “v-lookup” table function in Excel when the query (gene of interest in Column A) is not found in any of the clusters (tabs). We added a detailed description in revised tables.

5, Figure1 labeling does not seem to be complete the difference between Aa and Ab subpanels are unclear. It is hard to understand the differences between Aa and Ab

We revised the Figure (now 1A) and shortened the technical description

6, Figure 2. There is no scale shows for the bubble plots and it is not clear whether or not the same scale was used for multiple analysis. The upper and lower panel differences are not labelled.

Thanks for catching this. The two panels (E & F) that have another scale are now marked with an asterisk and the scale was added.

7, On figure 2 the authors make conclusion on the number of cells in each cluster, however the single cell dataset is relatively small and the single cell analysis suffers from uneven drop-out. I think claims made on cell fractions would need to be supported by an independent method. For example stain and quantification.

The discussion of Wnt4 served to raise two points:

1) To show that P0 cells producing more nephrons (Six2KI and Six2TSC) do not have a “younger” signature (Six2+, Cited1+, Wnt4–). This conclusion did not rely only on Wnt4. In fact, age was the main component separating three young samples from three old ones in UMAP under multiple filtering modes (Figure 2L)

2) That Wnt4+ cells constituted a greater fraction of the niche at P0. We confirmed the abundance of Wnt4 in the P0 Six2TSC relative to Six2TGC by bulk RNA sequencing on sorted NPC. Pseudo-bulk (Figure R6) and the violin plots (Figure R5) also suggest increased fraction of Wnt4 expressing NPCs at P0. However, RT-qPCR on sorted NPC detected a non-significant trend of higher Wnt4 and Cited1 in POTSC relative to TGC. RT-qPCR sums the total levels of Wnt4 in a population, whereas scRNA-seq suggested higher levels in some of the clusters comprising the population. Therefore, we are only comfortable at present stating that the P0 cells population does not resemble young cells and appear to contain a greater fraction of cells expressing higher levels Wnt4. We have changed the text to refrain from quantitative statements.

8, Another critical limitation of the single cell analysis is that batch and biological variations cannot be dissected as the batch represents both a technical and biological differences. This effect is particularly problematic for datasets such as the current work, where there no biological replicates were analyzed.

Although we do not explicitly have biological replicates for each single-cell genotype/time-point, since we have multiple samples for each time-point (n=3) and multiple samples for each genotype (n=2), we can assess what effects are specifically due to genotype or time-point, with these replicates. For example, we find TGC vs. TSC differences at P14 and P0 that are highly consistent (cell frequency, % of differential expressed genes). Similarly, we find consistent time-point differences (old versus young) when considering replicates (cell frequency, % of differential expressed genes – Figure 2). Moreover, these changes are robust over 6 different gene filtering options for the unsupervised analysis (Fig. 2L). We agree that we did not have biological replicates which are important in order to tackle batch effects. As we noted in our response to Reviewers 1 and 2, we have compared our clusters to an independent published dataset (Combes. et al) to address the difference between batches. As their public data had biological reps, we treated our data as if it

were another replicate and analyzed for batch effects using MNN. The robust agreement with their clusters helps us determine that our dataset did not suffer from severe batch effects.

Figure R4. Top 10 DE marker for each cell in our analysis. Note that *Cited1* is enriched in E14KI and P0TGC, and *Wnt4* is enriched in P0KI and P0TSC relative to control P0TGC.

I think statements made around *fig2N* etc must be confirmed by independent methods.

We agree, and thus have removed these panels from Figure 2.

9, If the primary goal was to understand differences between 3 genotypes why did not the author simply perform a cluster specific differential expression?

As the reviewers are all aware, one does not describe the litany of failures preceding success...We have performed differential gene expression analysis at an early stage of this project, both in bulk and pseudo-bulk RNA, and found it to be uninformative. To illustrate the point, we are including the heat map of top DE marker genes from the scRNA-seq (Figure R4). To generate this DE heat map, differential gene expression for each of the samples against a pool of the other five samples was identified by EDGER. The top 10 markers are shown in Figure R4. *Cited1* and *Wnt4* (red) are marked. While this provides some information similar in content to Figure 2, it is not providing any insight as to why *Six2*^{KI} and *Six2*^{TGC; Tsc1} have more nephrons.

The bulk RNA analysis was not informative either. Comparison of control P0TGC and P0TSC DEG that passed the FC and FDR cutoff identified *Wnt4* (elevated in P0 *Six2*TSC) but not *Cited1*.

10, The separation of the samples in the Velocito data is concerning and the authors should show convincing data that this is not a batch effect issue in here. Can this be recapitulated on a single sample? The authors should show separately *Six2*, *Cited* and *Wnt4* feature plots for the velocity data. From this data it seems *Hnf4a* is not really expressed. Again, one need to keep in mind that this is also batch effect sensitive. The arrows cannot be seen on the figure due to its size and resolution.

Efforts to validate Wnt4/Cited co-expression via RNAscope and flow cytometry did not support our original statements. Subsequent RNA-velocity analysis using updated dynamic models failed to recapitulate the initial suggested finding and thus this component of the manuscript has been removed.

11, Where is Tsc expressed in the single cell data? Where is its highest activity? Where is mTOR?

We added Figure R5 for the reviewer. Note that mTor and Tsc1 RNA might be missing from many cells due to dropout; largely a similar fraction of cells contains these transcripts in each sample. As for activity, staining for phospho-ribosomal protein S6 (refer to Figure 3 in the revised manuscript) and p-4EBP suggests overall mTor activity is low in NPC relative to proximal tubule epithelia and stroma. One can interpret this to suggest Tsc1 activity as an Mtor inhibitor is higher in NPC, but we think that the translation effect we report here may not be dependent entirely on mTor (Figure 3 in revised manuscript).

Figure R5. Feature UMAP plots showing the expression of *mTor*, and *Tsc1*, Note that age is separating the E14 samples from each other and the P0 samples along UMAP1 on the X axis, the P0 samples are separating from each other along UMAP2 on the Y axis.

12, I think the TRAP data is very nice, albeit I do not agree with the justification. Both scRNA and TRAP analyzes gene expression so TRAP will not be able provide insight on protein expression.

While it is true that TRAP still relies on RNA, studies confirmed it has a higher concordance with proteome than scRNA or bulk RNA sequences. Importantly, we show this concordance with the mRNA vs. protein or translome vs. proteins for *Fzd10* and *Lgr5* (Figure 4). Furthermore, we note that differential Wnt signaling components were identified only in the translome, and that one of two putative candidates (*Rspo3*) was demonstrated to contribute to the *Tsc1* phenotypes (delayed nephrogenesis cessation and increased NN). Analysis of the scRNA dataset using CellChat¹⁶ failed to reveal differential signaling mechanisms with NPC aging (Supplementary Figure 4), suggesting that use of TRAP (more representative of the proteome and not vulnerable to dropout of undetected transcripts) was more informative than scRNA-seq.

13, I think the Wnt data is interesting. The *Rspo3* data is interesting and it is nice to have the *foxd1* as a control dataset, however the wnt level is likely to be very tightly balanced so maybe low and high wnt levels could be problematic.

We agree, and we are not claiming that Wnt levels vary with age. Instead, we describe age-dependent change in cellular “perception” of a fixed (or slightly fluctuating) Wnt9b levels produced by the UB. We clarified this in the text. Out of curiosity we asked if Wnt9B mRNA levels were changing in time, using the UMAP analyses of UB cells from young vs old kidneys of the same genotype. We did not see any differences in Wnt9b expression level or distribution with age in the UB using this low resolution tool.

14, We would need to know results of the *Tsc Rspo3* het mice? Are these changes specific for *Tsc*?

Since we do not *a priori* expect synergy in this compound genotype, it was unclear to us how the *Tsc1*, *Rspo3* double mutant would be informative. We performed this experiment nonetheless and we find this

combination to be uninformative; nephron numbers did not display any synergy (Figure R6). We can envision the following scenario to explain the results:

- Suppose $Six2^{TGC}$; $Tsc1^{+/f}$; $Rspo3^{+/f}$ translated very few $Rspo3$ molecules, resembling NPC-specific $Rspo3$ nulls which are viable and fertile but are at the low end of $Six2^{TGC}$ range, perhaps by impairing NPC self-renewal/MET (as per ⁶). $Tsc1$ in this background rescues NN to the high end of the $Six2^{TGC}$ range.

Figure R6. All the offspring from the listed mating are compared to unrelated $Six2^{TGC}$; $Rspo3^{+/f}$ mice sharing a common genetic background. It is clear there is no synergy between $Tsc1$ and $Rspo3$. When the data are reorganized, the $Six2^{TGC}$; $Rspo3^{ff}$ mice have fewest nephrons, $Six2^{TGC}$; $Rspo3^{+/f}$ have most, the $Six2^{TGC}; Tsc1^{+/f}; Rspo3^{+/f}$ kidneys are at an intermediate level.

15, The $Tmem59$ data is clearly the highlight of the manuscript. How does Tsc regulate $Tmem59$? Where is $tmem59$ expressed? How does $Tmem59$ lead to an increase in nephron number?

$Tmem59$ is expressed quite ubiquitously across cell types, tissues and timepoints. Based on the transcriptome data we leaned on the literature⁹ suggesting that $Tmem59$ creates a Wnt signalosome by clustering FZD receptors, increasing avidity to Wnt9b ligand, and making it a candidate for further analysis. Importantly, in our original submission we overlooked the fact that $Tmem59$ was reduced, not lost, in $Tsc1$ hemizygote NPC. Therefore, we asked in the revised manuscript if heterozygosity for $Tmem59$ (which does not enhance NN) synergizes with $Rspo3$. Synergy and cessation timing analyses suggest that $Tmem59$ heterozygosity is not a contributor to the $Tsc1$ phenotype and thus no longer a focus of this paper (Supplementary Figure 5H) and delegated much of this story to supplementary information. While the gain of nephrons in homozygous $Tmem59$ mutants is immaterial to the mechanism of action downstream of $Tsc1$, it is consistent with the model suggesting that tuning Wnt reception will increase NN. Future studies investigating the mechanism by which homozygosity for $Tmem59$ mutation can alter NN are not within the scope of this

manuscript. In its place we considered if metabolism/mTor contribute to Tsc1 activity and show they are unlikely to do so. The mechanism through which hamartin dosage impacts translation of Tmem59 and other proteins is beyond the scope of the current study. It is the subject of a major ongoing effort in the lab, but it is at an early, uninformative stage.

Supplementary Figure 5: (H) Nephron counts in adult (\geq P28) mice determined via the acid maceration method for glomerular counts. Two-tailed unpaired t tests were performed in GraphPad Prism version 8 to evaluate statistical significance of single kidney nephron numbers. Both kidneys were counted for each animal with males and females represented. *Tmem59* reduction does not synergize with *Rspo3*.

16, The resolution of the figures are too low and the labels cannot be read on figure1

All figures are now all at 300dpi. Figure 1 has been revised and presented more clearly.

Reviewer #4 (Remarks to the Author):

The authors in this study take advantage of genic animal models, mainly RNA-seq and bioinformatics tools to understand the mechanisms of nephrogenesis. The tipping-point hypothesis is intriguing but more evidence is needed to prove it, and more evidence is also needed to support their conclusions in the discussion.

Comments:

- The translatoome, which is different from transcriptome, is a more accurate approximation for estimating the expression level of some genes. The advantage of TRAP technique designed for animal studies is that TRAP allows affinity purification of ribosome-bound mRNA directly from the- targeted tissues for further analysis of gene expression and identification of new transcriptomes and translatoomes. This may avoid potential artifacts resulting from tissue fixation, dissociation of tissues, or isolation of single cells from tissues when the entire translated mRNA complement of specific tissues is examined from intact tissues in vivo. Thereby, if the isolated cells from transgenic animals as described in this manuscript are used for TRAP and RNA-seq, the value will be decreased or limited due to the changes in the ribosome-bound RNA resulting from the isolation process.

While we agree that performing a one-step TRAP is preferable and superior method, and that cell isolation may be a confounder, we note that in the literature some degree of cell isolation preceding TRAP is practiced (e.g.,^{17,18}). Having analyzed the Notch2 BACTRAP line and finding no expression in NPC (see Figure R1 above), we produced the Cited1-TRAP line, but RPL10a-eGFP expression was too low to facilitate direct isolation of polysomes from this line as well. Since Six2 persists in PTA, we did not try to drive RPL10a-eGFP from this enhancer as a sorting step would have been necessary as well. We decided instead to use Six2^{TGC} followed by separation of NPC progenitors from RPL10a-eGFP-expressing descendants by FACS. We point out that throughout the isolation and sorting (all done on ice) translation was inhibited with cycloheximide, which traps elongating ribosomes, and that the methodology was the same for *Tsc1*^{+/-} and control NPC. That means that we may have lost a few ribosome-associated transcripts. To confirm our key findings, and control for the possibility these results were influenced by the isolation step, we did analyze the expression of differentially translated proteins and show reduction in abundance. We more clearly discuss the cofounders in the revised text.

- It is not clear how the studies in result section 1 support the statement "The cellular composition of the nephron progenitor niche is unaffected by mutations ...",

See response to Reviewers 1 and 3. In short, we confirm that the same cell clusters seen by others in the community are present in our analyses of these mutants, and that the genetic background differences did not result in significant enough changes to gene expression to split a cluster of NPC into a sub-cluster with markers distinct from the ones described before. Variation in clustering also did not alter the overall

assignment of cellular fates, attesting to the robustness of the clustering. We have however improved the presentation of this section.

- More biological evidence is needed to support the "tipping point" hypothesis. General description that multiple pathways and mechanisms may be responsible for the gene expression differences that separate the indicated genotypes appears not sufficient as a suggestion. What possible pathway and mechanisms are involved? Could the authors show the phenotype evidence of different stages for the mice with different genotypes?

The biological evidence for the tipping point hypothesis were presented in an earlier study, by Chen et al⁸. In that study, we explored prediction of models stipulating purely intrinsic or purely extrinsic mechanisms controlling progenitor lifespan using a progenitor transplantation assay that allowed us to compare engraftment of old and young progenitors into the same young niche. The results provided the basis for formulating the tipping point model in which intrinsic age-dependent changes affect inter-progenitor interactions that drive cessation of nephrogenesis: The progenitors displayed an age-dependent decrease in proliferation and concomitant increase in niche exit rates. Importantly, 30% of old progenitors remained in the niche for up to a week post-engraftment, a net gain of 50% to their lifespan, but only if surrounded by young, Fgf20-expressing neighbors.

- It could be helpful to provide cellular and molecular evidence that cell population switches from self-renewal to differentiation.

Several groups have demonstrated this, including the McMahon and Carroll groups (for example, see¹⁻³). Our contribution is to show that the levels of ligands (a change in the environment contributed by the UB) need not vary with age. Instead, a change in the ability of NPC to both modify the environment (by translating more RSPO proteins, producing less FGF20) and their perception of it (more Wnt antagonists) can delay NPC niche exit

Minor

concerns:

1. In lines 189-198, authors should discuss these changes in more detail. Furthermore, further discussion on the differences between the rest of the genes analyzed should be added.

We assume the request is addressing Figure 2. We clarified this description to focus on the conclusion that the $Tsc1^{+/-}$ NPCs are not more youthful in their transcriptional signature based on expression of markers along the mesenchymal-to-epithelial lineage progression. We also added violin plots (Supplementary Figure 2) to better illustrate cluster-specific expression of these marker genes.

2. What is reason why the authors decided not to evaluate the Six2KI genotype for transcriptome analysis (TRAP). In line 245 authors mention that "the increased nephron endowment in Six2KI might be driven by a mechanism acting earlier in nephrogenesis, before P0". It would be interesting to verify with this genotype if there is any change in the transcriptome with respect to the other two genotypes.

It would be, but given that our Six2KI line ($Six2^{+/CE}$) would require tamoxifen to activate Cre and turn the expression of EGFP-Rpl10a for TRAP, we felt the timing and efficiency of tamoxifen administration would present confounders complicating the comparison to controls.

3. In the introduction section, authors should mention a little more about Wnt pathway and its relation to the nephron endowment/nephrogenesis. Since the authors focus most of the results in this pathway.

We have augmented the introduction as requested.

4. In line 50 "nephrogenesis ends by P3", the meaning of P3 should be added.

Postnatal day 3 (P3) was added.

5. In line 64 “and can and contribute nephrons”, please remove the second word “and” from the sentence.

Thanks! Done.

6. In line 78 “transcriptome at two developmental timepoints (E14 and P0) of three genotypes”, the meaning of E14 and P0 should be added.

Embryonic day 14 (E14) and postnatal day 0 (P0) have been defined accordingly.

7. In lines 139 and 148 specific reference should be added.

Thanks! Done.

8. In Fig 1B and Fig 2M, the image resolution should be improved.

We have improved the figures as requested.

9. Reference to Fig 3C is missing in the text, please include it where it belongs.

We note that this figure is not included in the revised manuscript.

10. In line 495, the concentration of RNase inhibitors and protease inhibitors should be included.

Thanks! We added the concentrations.

11. In line 531, please complete the reference with the name of the authors.

Alison Jarmas performed the counts, her initials were added.

12. The authors studied E14 and P0 but in the method section for protein quantification they stated that kidneys from E14 or P0, P1, or P2 mice were isolated in ice-cold PBS and the capsule removed.

Figure 4 and the methods section have been updated to clarify that P0, P1 and P2 mice were used for protein analysis via flow cytometry.

References

- 1 Park, J. S., Valerius, M. T. & McMahon, A. P. Wnt/{beta}-catenin signaling regulates nephron induction during mouse kidney development. *Development* **134**, 2533-2539 (2007).
- 2 Ramalingam, H. *et al.* Disparate levels of beta-catenin activity determine nephron progenitor cell fate. *Dev Biol* **440**, 13-21, doi:10.1016/j.ydbio.2018.04.020 (2018).
- 3 Karner, C. M. *et al.* Canonical Wnt9b signaling balances progenitor cell expansion and differentiation during kidney development. *Development* **138**, 1247-1257, doi:dev.057646 [pii]10.1242/dev.057646 (2011).
- 4 Kuure, S., Popsueva, A., Jakobson, M., Sainio, K. & H., S. Glycogen Synthase Kinase-3 Inactivation and Stabilization of β -Catenin Induce Nephron Differentiation in Isolated Mouse and Rat Kidney Mesenchymes. *J Am Soc Nephrol* **Published on line as doi: 10.1681/ASN.2006111206** (2007).
- 5 Lindstrom, N. O. *et al.* Integrated beta-catenin, BMP, PTEN, and Notch signalling patterns the nephron. *eLife* **3**, e04000, doi:10.7554/eLife.04000 (2015).

- 6 Vidal, V. P. *et al.* R-spondin signalling is essential for the maintenance and differentiation of mouse
nephron progenitors. *eLife* **9**, doi:10.7554/eLife.53895 (2020).
- 7 Li, Z. *et al.* 3D Culture Supports Long-Term Expansion of Mouse and Human Nephrogenic
Progenitors. *Cell Stem Cell* **19**, 516-529, doi:10.1016/j.stem.2016.07.016 (2016).
- 8 Chen, S. *et al.* Intrinsic Age-Dependent Changes and Cell-Cell Contacts Regulate Nephron Progenitor
Lifespan. *Dev Cell* **35**, 49-62, doi:10.1016/j.devcel.2015.09.009 (2015).
- 9 Gerlach, J. P. *et al.* TMEM59 potentiates Wnt signaling by promoting signalosome formation. *Proc
Natl Acad Sci U S A* **115**, E3996-E4005, doi:10.1073/pnas.1721321115 (2018).
- 10 Short, K. M. *et al.* Global quantification of tissue dynamics in the developing mouse kidney. *Dev Cell*
29, 188-202, doi:10.1016/j.devcel.2014.02.017 (2014).
- 11 Goentoro, L. & Kirschner, M. W. Evidence that fold-change, and not absolute level, of beta-catenin
dictates Wnt signaling. *Mol Cell* **36**, 872-884, doi:10.1016/j.molcel.2009.11.017 (2009).
- 12 Volovelsky, O. *et al.* Hamartin regulates cessation of mouse nephrogenesis independently of Mtor.
Proc Natl Acad Sci U S A **115**, 5998-6003, doi:10.1073/pnas.1712955115 (2018).
- 13 Kristensen, A. R., Gsponer, J. & Foster, L. J. Protein synthesis rate is the predominant regulator of
protein expression during differentiation. *Mol Syst Biol* **9**, 689, doi:10.1038/msb.2013.47 (2013).
- 14 Doyle, J. P. *et al.* Application of a translational profiling approach for the comparative analysis of CNS
cell types. *Cell* **135**, 749-762, doi:S0092-8674(08)01366-4 [pii] 10.1016/j.cell.2008.10.029 (2008).
- 15 Miyazaki, J. *et al.* Expression vector system based on the chicken beta-actin promoter directs efficient
production of interleukin-5. *Gene* **79**, 269-277, doi:10.1016/0378-1119(89)90209-6 (1989).
- 16 Jin, S. *et al.* Inference and analysis of cell-cell communication using CellChat. *Nat Commun* **12**, 1088,
doi:10.1038/s41467-021-21246-9 (2021).
- 17 Megat, S. *et al.* Differences between Dorsal Root and Trigeminal Ganglion Nociceptors in Mice
Revealed by Translational Profiling. *J Neurosci* **39**, 6829-6847, doi:10.1523/JNEUROSCI.2663-
18.2019 (2019).
- 18 Mansuri, M. S. *et al.* Differential Protein Expression in Striatal D1- and D2-Dopamine Receptor-
Expressing Medium Spiny Neurons. *Proteomes* **8**, doi:10.3390/proteomes8040027 (2020).

REVIEWER COMMENTS

Reviewer #1 (Remarks to the Author):

The authors did a very thorough job on the new version of this manuscript. All of my previous concerns have been addressed satisfactory.

I agree with the reasons why the authors say they cannot get more data on the effect on beta-catenin signaling, indeed better tools are urgently required, but simply not available at the moment. They did what they could in this respect, what more could a reviewer ask for..?

I'm glad the authors agreed with my criticism on the Tmem59 mouse model as described in the original manuscript. It's a pity the original idea was found not to be correct, but they way this is analyzed now is complete and worthy of publication.

I simply love the inclusion of the data from the collaborative cross. This is an enormous powerful resource for these sort of things where many people don't even know it existence, let alone what it can be used for...

Finally I thanks the authors for the lesson in heterozygous/hemizygous nomenclature, I will have to update the slides I use for teaching mouse genetics, Life-long learning indeed :-)

For me I fully support accepting this manuscript for publication despite the original idea on the role of Tmem59 not being correct. I do not think that this makes the manuscript less interesting, the science is just as good whether the hypothesis is correct or not. However this is a final call for the editor to make.

Reviewer #3 (Remarks to the Author):

Overall it seems that the authors had to make very drastic changes to the manuscript as new experiments did not support their earlier claims.

This is very unfortunate and the presented work fails to provide critical novel insights into kidney development.

Just a few hopefully helpful suggestions.

1. The PCA plot is usually used to determine optimal number of cell clusters in the single cell data
2. Removing ribosomal and mitochondrial genes could significantly alter the results of the single cell data analysis
3. Gene filtering methods can not really substitute for replicates
4. While pooling samples might help reducing variability, the overall conclusion is limited as only a single sample per condition is analyzed precluding important statistical analysis
5. Good portion of statements made in during the initial submission has been removed, including Fig2N, RNA velocity analysis.
6. In general feature plots (response 11) is not a quantitative method to characterize expression
7. Lots of speculations were made around Rspan3, however the results do not support this initial speculation (response 14)
8. More details work did not seem to support the role of Tmem59 (response 15)
9. Finally and most importantly the authors failed to see gene expression differences between the 3 different genotypes in the single cell data (response 9)

Reviewer #4 (Remarks to the Author):

The authors have addressed my concerns.

We thank the editorial team and each of the reviewers for their time in evaluating our revised manuscript. We are pleased that Reviewer #1 commented that our resubmission represents a “very thorough” effort worthy of publication and that Reviewer #4 is similarly satisfied with our efforts to address their concerns as well.

Regarding the additional concerns raised by Reviewer #3, please find below our detailed response, first discussing the broader question of novelty of the work, and then addressing the specific comments, which primarily relate to aspects of the single-cell analysis. Note here the addition of an author, Dr. Nathan Salomonis. Nathan was our consultant throughout but his contributions to the latest analysis and presentation of the single-cell data in our newly revised manuscript warrant his inclusion as an author:

Reviewer #3 (Remarks to the Author):

Overall, it seems that the authors had to make very drastic changes to the manuscript as new experiments did not support their earlier claims.

This is very unfortunate and the presented work fails to provide critical novel insights into kidney development.

We ask that the reviewer and editors consider the findings we present, which have been strengthened by the review process as described below:

What we report in both versions of the manuscript:

1. We show, via RNA-Seq of ribosomal protein L10a-associated transcripts (translatome), that $\text{Six2}^{\text{TGC};Tsc1}$ (herein $Tsc1^{+/-}$) produces *selective* differential post-transcriptional changes, notably in *Rspo3* (these changes are not apparent, importantly, in the transcriptome as confirmed by RT-qPCR of FACS-isolated NPCs across 5 replicates, or in DEG analysis of the NPC clusters, added as Supplementary Table S7, and Supplementary Figure 1I).
2. Translatome data provides mechanistic insight. Analysis of the translatome data identified 6 prioritized candidates. We elected to pursue *Rspo3* and *Tmem59*; deletion of one (*Tmem59*) and reduction in another (*Rspo3*) both led to increase in nephron number.
3. We identify *Tmem59* as a novel regulator of nephron number. Note that because *Tmem59* is not lost in $Tsc1^{+/-}$, and reduction in *Tmem59* does not elevate nephron number, it does not explain the nephron gain observed in $Tsc1^{+/-}$.
4. By contrast, reduction in *Rspo3* recapitulates and can explain mechanistically the $Tsc1^{+/-}$ phenotype.
5. Stromal cells are responsive to genetic manipulation of NPCs.

New supportive data added:

1. We observed delayed cessation of nephrogenesis in *Rspo3* hemizygotes, consistent with our proposed model, first observed by us in the $Tsc1^{+/-}$ pups (Volovelsky et al., 2018).
2. Independent replicate (n=5) testing of RNA transcripts by RT-qPCR did not detect transcriptional level differences in *Rspo3* or other Wnt-related mRNAs evaluated in the translatome/via flow cytometry (*Fzd10*, *Lgr5*). scRNA analysis at any depth should not supplant these PCR-based results, when considering rigor.

3. Early scRNA analysis directed our attention to translation (Chen et al., 2015), which led us to investigate the mTor pathway and its role in cessation (Volovelsky et al., 2018), which in part motivated our analysis of the translome here (this direction is further supported by the scRNA analysis presented here, which we clarify in our newly revised manuscript). We offer new evidence of independence of this translation-level mechanism, mediated by *Tsc1*, from changes in global mTORC1 activity in nephron progenitor cells. This is a compelling finding, and may have relevance to other cell types in which *Tsc1* can alter translation with a specificity previously unknown.
4. We have more deeply examined the role of mitochondrial respiration, another hypothesis emerging from scRNA findings reported previously (Chen et al., 2015) and herein; we show it is not differentially altered in *Tsc1*^{+/-} NPC and thus cannot be the underlying mechanism for delayed timing of cessation in this model (see below).

In aggregate we feel the strengths and novelty of our findings were not eroded during the revision; we show that *Tsc1* selectively alters post-transcriptional levels of *Rspo3*, a Wnt pathway agonist, which modulates nephrogenesis cessation timing and subsequently, nephron number. Collectively this provides molecular detail to a plausible tipping point mechanism for nephron progenitor cell niche exit.

We appreciate the reviewer's additional comments on technical aspects of the scRNA-Seq analysis; we have made the indicated changes to improve the thoroughness and statistical rigor of our presented analysis.

In broad terms, the scRNA-Seq results were informative and in concordance with earlier work (Chen et al., 2015); we validated changes associated with NPC aging in the ribosomal and mitochondrial/metabolic signatures. These global signatures prompted us to 1) pursue TRAP analysis (translatome) and 2) address the hypothesis that a metabolic mechanism contributed to the phenotype. While metabolism switches from glycolysis to respiration as NPCs near niche exit (Liu et al., 2017), we show that these same changes occur in *Tsc1*^{+/-} mutants as in controls by postnatal day 0 and therefore are unlikely to contribute to the extension of nephrogenesis.

We clarify that our goal in this manuscript is to develop and test a hypothesis for how *Tsc1* hemizyosity elevates nephron number as compared to the *Six2*^{TGC} control. From the scRNA-Seq analyses conducted in this study, analysis of differentially expressed genes (DEGs, see response #9 below and now added to our manuscript) revealed two signatures (translation and metabolism), which motivated the subsequent studies presented in our manuscript.

As we only generated one dataset per genotype/time, we readily acknowledge that additional replicates may reveal additional, transcriptional-level mechanisms that could form the basis of a future study, but does not impact the central premise and conclusion we present in this manuscript. In our new revisions, we leveraged the fact that we pooled cells from multiple animals to separate male, female and undetermined cells into pseudo bulk pools to address some of the concern raised by lack of replicates (see response #4 below). We also provide a refined set of differentially expressed genes (DEGs; see response #9 below):

Please find our responses to each specific point discussed below:

Just a few hopefully helpful suggestions.

1. The PCA plot is usually used to determine optimal number of cell clusters in the single cell data

As part of the standard Seurat workflow, we performed PCA which suggested the first 20 PC will be informative and identified a conservative resolution of 0.8. As we showed in our earlier response, we varied the resolution between 0.5 and 1. We now further provide principal component and T-SNE plots for each of the results from different resolutions in the included GitHub repository for this manuscript (<https://github.com/praneet1988/Age-Dependent-Changes-in-the-Progenitor-Translatome-Coordinated-in-part-by-Tsc1-Increase-Perception->).

2. Removing ribosomal and mitochondrial genes could significantly alter the results of the single cell data analysis

We agree, as demonstrated in Figure 2L. For our subsequent analyses, we included all of these transcripts.

3. Gene filtering methods can not really substitute for replicates

We agree; see response #4 for a discussion of replicate analyses.

4. While pooling samples might help reducing variability, the overall conclusion is limited as only a single sample per condition is analyzed precluding important statistical analysis

A. Since each of our samples represents cells pooled from multiple animals' kidneys, we split the cells into male, female and undetermined within each sample for each cell type on the basis of the ratio of female and male-specific transcripts (<https://git.io/Jlevy>). This produces "pseudo-bulks" profiles for each cell type, separated into male, female and unassigned for each capture, providing sex-based replicates for differential analyses. We find consistency of such pseudo-bulks across all models, when considering best marker genes (Figure R1 and Supplementary Figure 1H in the revised manuscript, Supplementary Table S6; each column on lower x-axis represents a unique cell type, age/genotype and assigned sex combination).

MarkerGenes_correlations-ReplicateBased

Figure R1. MarkerFinder analysis (AltAnalyze) of all cell types assigned male, female or unassigned reveals consistent gene expression patterns across these biological pseudo-replicates.

B. Considering the sex-based pseudo-bulk samples for each cell-type as replicates, we were able to perform a holistic differential expression analyses for all cell populations using the previously published cellHarmony workflow in the software AltAnalyze (fold > 1.2, empirical Bayes-moderated t-test $p < 0.05$). This orthogonal analysis does not consider individual cells, but rather the union of cells from males, females or unassigned, as an independent statistical assessment. cellHarmony identifies pathway enrichment results and individual differentially expressed genes that largely match those in our prior cell-based analyses. We performed direct and indirect comparisons of E14 KI vs. TGC and E14 KI vs. TSC, to demonstrate consistent and divergent changes between these two comparison groups along with corresponding statistically enriched pathways. In addition, we now contrast pseudo-bulk replicate differentials between TSC and TGC at P0 (Supplementary Figure 1H).

C. We have leveraged independent, published data for an age-and-genotype matched sample and found that our sample was consistent with those replicates. This was/is discussed in the manuscript and previous reviewer response.

We mentioned one of these remedies (C) in our previous response and included them in our revised manuscript to establish that all our samples are within the range (in cluster number, marker identity, and gene expression) of the published datasets. In each case, whichever method we used, we find a high degree of consistency in our results.

5. Good portion of statements made in during the initial submission has been removed, including Fig2N, RNA velocity analysis.

We appreciate the reviewer's prior note that methods such as RNA Velocity may be not be reliable. There is no substitute for definitive experimental validation; in this case, RNAScope for *Cited1* and *Wnt4* transcripts did not support the *in-silico* predictions. Furthermore, subsequent updated bioinformatics packages did not recapitulate the initial observation. Therefore, this aspect of the manuscript was removed, and our secondary hypothesis that *Wnt4*-expressing progenitors migrate back to the naïve state to a greater extent in *Tsc1*^{+/-} was discarded. The overall conclusions remained unchanged.

6. In general feature plots (response 11) is not a quantitative method to characterize expression

Response 11 relates to characterizing where mTOR and Tsc1 are expressed, using feature plots. It is accepted that these genes are ubiquitously expressed (see GTEX data (human) below, and lookup BioGPS data for the mouse). We added here violin plots (Figure R2) for these genes in our samples by cell type below the GTEX data.

Figure R2. Gene expression violin plots for Mtor and Tsc1 across human tissue, mouse tissue and the

kidney cell types presented in our scRNA-Seq datasets.

7. Lots of speculations were made around *Rspo3*, however the results do not support this initial speculation (response 14)

We emphasize that both our initial and revised manuscript support the role we propose for *Rspo3* in regulating nephron number. In the revised manuscript, we added evidence that delayed cessation timing is also occurring in *Rspo3*^{+/-} kidneys, strengthening the link to the *Tsc1*^{+/-} phenotype. No change in cessation timing is observed in *Tmem59* KO, again consistent with a role independent from *Tsc1* but consistent with a possible function in Wnt signaling, if *Tmem59* potentiates a Wnt signalosome as proposed in the literature (Gerlach et al., 2018).

Response 14 addressed the reviewer's hypothesis that *Tsc1* and *Rspo3* might synergize. As we stated, lack of synergy would not invalidate our interpretation. For synergy to occur, reducing *Rspo3* further (caused by combining hemizygoty with the translational suppression in *Tsc1*^{+/-}) should have a benefit. As reported (Vidal et al., 2020), reducing the levels of *Rspo3* below a certain threshold is detrimental to nephron generation, likely causing impaired self-renewal and reduced MET. What we observed instead of synergy was that *Tsc1/Rspo3* compound hemizygotes displayed an intermediate nephron number phenotype, less than in *Tsc1* or *Rspo3* single gene hemizygotes but importantly, greater than reported in *Rspo3* nulls. The parsimonious interpretation of this result is that indeed, reducing *Rspo3* below the hemizygote levels is deleterious. This result provides validation that reduction in hamartin selectively impairs *Rspo3* translation.

Finally, we remind the reviewer here that our model of interest, *Tsc1*, also exhibits dose-dependent effects in which hemizygoty is beneficial for nephron generation but further reduction, as in the null state, is detrimental due to the emergence of other pathologies.

8. More details work did not seem to support the role of *Tmem59* (response 15)

Based on bioinformatics analysis of the TRAP dataset, we identified 6 candidates of which we elected to pursue two. While we no longer suggest a role for *Tmem59* in mediating the *Tsc1*^{+/-} phenotype, we validated our initial report that *Tmem59* can modulate nephron number with genetically purified lines, a significant novel finding even if not illuminating the mechanism of *Tsc1* action.

9. Finally and most importantly the authors failed to see gene expression differences between the 3 different genotypes in the single cell data (response 9)

We clarify here our previous response (response 9) by providing additional details of our scRNA-Seq analysis. We do identify gene expression differences between the three genotypes and the two timepoints analyzed in the single cell studies. We now include this data in the revised manuscript as new Supplementary Table S7. We also clarify the manuscript text stating that the DEGs identified, and subsequent functional enrichment analyses with ToppFun (<https://toppgene.cchmc.org/enrichment.jsp>) concurred with our previous data that initiated our pursuit of translational and metabolic mechanisms. Furthermore, the lack of orthogonal validation (by RT-qPCR and RNAScope) for differentially expressed genes between the three samples (*Wnt4*, *Cited1*), for which we did develop a mechanistic hypothesis in combination with RNA Velocity in the original submission, reduced our confidence in drawing single gene-level conclusions, in part due to lack of replicates and in part due to the technique's vulnerability to significant dropout of transcripts, and the dependence of the final DEG set on upstream steps,

e.g., normalization, mapping, etc'. These facts furthered our enthusiasm for the bulk-RNA level TRAP approach, performed with biological replicates.

We augment our discussion of differential gene expression for the reviewer as follows:

1. We provide all the data for differential gene expression between samples with associated fold changes and significance/statistics. We added single cell barcode to cell cluster association as Supplementary Table S4 so that future readers can revisit this data.
2. As an orthogonal assessment, we include analyses of sex-based pseudobulk replicate differentials derived from each captures using the software cellHarmony, presented as new Supplementary Figure 1H and Supplementary Table S6.

The inclusion of orthogonal replicate analyses, based on pseudobulks rather than individual cells, provides additional statistical strength to the predictions. We find a high degree concordance the single-cell differentials for each cell-type and sex-based pseudobulk differentials (see revised Supplementary Table S7). Our focused independent experimental validations point clearly to consistent transcriptome impacts but do not definitively negate these transcriptome findings which we were not able to experimentally verify in these studies. Nonetheless, this does not alter our primary findings.

In summary, we have revised our manuscript to better communicate both the limitations and findings of our scRNA-Seq analysis, provide complete representation of the identified changes in gene expression, and emphasize the novelty of our findings, including *Tmem59* as a new negative regulator of nephron number and selective differential translation mediated by *Tsc1*. We thank Reviewer #3 for helping us improve the technical description and presentation of the scRNA-Seq analysis.

References

- Chen, S., Brunskill, E.W., Potter, S.S., Dexheimer, P.J., Salomonis, N., Aronow, B.J., Hong, C.I., Zhang, T., and Kopan, R. (2015). Intrinsic Age-Dependent Changes and Cell-Cell Contacts Regulate Nephron Progenitor Lifespan. *Dev Cell* 35, 49-62.
- Gerlach, J.P., Jordens, I., Tauriello, D.V.F., van 't Land-Kuper, I., Bugter, J.M., Noordstra, I., van der Kooij, J., Low, T.Y., Pimentel-Muinos, F.X., Xanthakis, D., *et al.* (2018). TMEM59 potentiates Wnt signaling by promoting signalosome formation. *Proc Natl Acad Sci U S A* 115, E3996-E4005.
- Liu, J., Edgington-Giordano, F., Dugas, C., Abrams, A., Katakam, P., Satou, R., and Saifudeen, Z. (2017). Regulation of Nephron Progenitor Cell Self-Renewal by Intermediary Metabolism. *J Am Soc Nephrol* 28, 3323-3335.
- Vidal, V.P., Jian Motamedi, F., Rekima, S., Gregoire, E.P., Szenker-Ravi, E., Leushacke, M., Reversade, B., Chaboissier, M.C., and Schedl, A. (2020). R-spondin signalling is essential for the maintenance and differentiation of mouse nephron progenitors. *Elife* 9.
- Volovelsky, O., Nguyen, T., Jarmas, A.E., Combes, A.N., Wilson, S.B., Little, M.H., Witte, D.P., Brunskill, E.W., and Kopan, R. (2018). Hamartin regulates cessation of mouse nephrogenesis independently of Mtor. *Proc Natl Acad Sci U S A* 115, 5998-6003.

REVIEWERS' COMMENTS

Reviewer #3 (Remarks to the Author):

I have no additional concerns with the presented work